# German funders' data sharing policies—A qualitative interview study

**Michael Anger** *, **Christian Wendelborn, Christoph Schickhardt**

Section for Translational Medical Ethics, Clinical Cooperation Unit Applied Tumor Immunity, National Center for Tumor Diseases (NCT) Heidelberg, German Cancer Research Center (DKFZ), Heidelberg, Germany

* michael.anger@kit.edu

## Abstract

### Background

Data sharing is commonly seen as beneficial for science but is not yet common practice. Research funding agencies are known to play a key role in promoting data sharing, but German funders' data sharing policies appear to lag behind in international comparison. This study aims to answer the question of how German data sharing experts inside and outside funding agencies perceive and evaluate German funders' data sharing policies and overall efforts to promote data sharing.

### Methods

This study is based on sixteen guided expert interviews with representatives of German funders and German research data experts from stakeholder organisations, who shared their perceptions of German' funders efforts to promote data sharing. By applying the method of qualitative content analysis to our interview data, we categorise and describe noteworthy aspects of the German data sharing policy landscape and illustrate our findings with interview passages.

### Results

We present our findings in five sections to distinguish our interviewees' perceptions on a) the status quo of German funders' data sharing policies, b) the role of funders in promoting data sharing, c) current and potential measures by funders to promote data sharing, d) general barriers to those measures, and e) the implementation of more binding data sharing requirements.

### Discussion and conclusion

Although funders are perceived to be important promoters and facilitators of data sharing throughout our interviews, only few German funding agencies have data sharing policies in place. Several interviewees stated that funders could do more, for example by providing incentives for data sharing or by introducing more concrete policies. Our interviews suggest the academic freedom of grantees is widely perceived as an obstacle for German funders in introducing mandatory data sharing requirements. However, some interviewees stated that

**Data Availability Statement:** This article reports findings from expert interviews with employees of funding agencies and stakeholder organisations. The full transcripts of the interviews contain highly sensitive qualitative data – including person-related

data – and are therefore subject to data protection law, particularly the EU-GDPR. Consent for participation in the interview study was obtained on the condition of complete anonymity and confidentiality. The interviewees' trust in strict anonymity and confidentiality allowed us to gain specific insights that constitute the value of our data (such as information and opinions about their work and the internal situation of organisations which are also their employers), but at the same time make an unlimited publication of the interview transcripts impossible for legal and ethical reasons. To enable replication of the findings reported in this article without violating our legal and ethical obligations, we created excerpts of the interviews. Since we cannot fully exclude the possibility that even the excerpts might still contain some information that could technically allow third parties to re-identify the interviewees, we grant access to the data solely upon request. This minimal data set is deposited in the heiDATA repository, DOI: https://doi.org/10.11588/data/FJAG0X. Ethical clearance was obtained – under the above-mentioned data protection conditions – from the Data Protection Office of the German Cancer Research Center, Heidelberg (registration no. D120-P989). Requests for data use and access will be reviewed and decided upon by the independent Data Use and Access Committee of the Heidelberg University Hospital. Please address data access and usage requests to use-and-access@med.uni-heidelberg.de.

**Funding:** The joint research project DATABLIC – Carrots and Sticks? Data Sharing Policies for (German) Public Research Funders. Ethical, Legal, Social and Behavioural Aspects is funded by the German Ministry of Education and Research (Bundesministerium für Bildung und Forschung, funding reference no. 01GP1904A). The funder had no role in study design, data collection and analysis, decision to publish, or preparation of the manuscript.

**Competing interests:** The authors do not have any competing interests. This study is part of a research project which is funded by the German Federal Ministry of Education and Research and reports to the project execution organisation DLR Projektträger. Both organisations were part of our larger field of study (German funders) but had no influence on the design or content of this study. This does not alter our adherence to PLOS ONE policies on sharing data and materials.

stricter data sharing requirements could be justified if data sharing is a part of good scientific practice.

## 1. Introduction

The sharing of research data for the purposes of reuse and reproducibility is gaining momentum and receiving increased attention, as can be observed both in scientific literature [1–8] and political efforts [9–12]. On the one hand, more and more actors in science emphasise the benefits of data sharing, referring to improved efficiency, reproducibility and transparency of scientific research due to better data availability [3–5, 13–18]. Data sharing is increasingly perceived as a part of good scientific practice [19–21]. On the other hand, and despite various efforts to encourage and promote it, data sharing is still not common practice [7, 15, 16, 22–26]. Data sharing is a complex topic and advancing it requires coordinated efforts by different stakeholders, such as funding agencies, journals, research infrastructures and different scientific communities [27–30].

It is a common notion that funders in particular can and should play a key role in promoting data sharing, as they provide critical funding for research and guidance through their data sharing policies [6, 7, 16, 28, 31–37] [Footnote: In this paper, we use the term "data sharing policies" in a broad sense and to describe a framework of grant conditions, instructions, incentives, evaluation mechanisms and monitoring criteria regarding data management and sharing. In practice, there are different stages of policy development, but we would expect an advanced data sharing policy to encompass all these aspects to some degree]. Several leading international funding agencies have introduced explicit policies on data sharing or FAIR (Findable, Accessible, Interoperable, Reusable) data [38] to encourage and advance data sharing [33, 35, 37, 39]. However, and although the German funding landscape is among the largest in the world [40, 41], a closer look at funding agencies in Germany reveals that data sharing policies are neither very advanced nor very widespread here [42]. The question arises: why could this the be case? Some studies already indicate that funders face challenges and have room for improvement when it comes to implementing and designing data sharing policies [25, 35, 39, 43–48]. Crucially, however, German funders' perceptions themselves remain widely underexplored even in literature on funding arrangements in Germany [25, 42, 48–53], let alone their perceptions of data sharing policies in Germany.

Drawing on a series of guided expert interviews with representatives of German funding agencies and German research data experts from other organisations, our article provides an account of the perceived status quo of German funders' data sharing policies, their efforts to promote data sharing and prevailing hurdles in advancing data sharing. In doing so, this study aims to answer the following research question: *How do data sharing experts inside and outside German funding agencies perceive and evaluate German funders' data sharing policies and overall efforts to promote data sharing*?

After providing some background on the German funder and data sharing landscape (chapter 2), we describe the socio-empirical methods of our study (chapter 3) and present the main findings from our interviews (chapter 4). We present our empirical data in five result sections: The interviewees' perspectives on the status quo of German funders' data sharing policies (4.1), their perspective on the role of funders in promoting data sharing (4.2), their perspectives on current and potential measures by funders to promote data sharing (4.3), their perspectives on general barriers and constraints for German funders (4.4) and their

perspectives on the implementation of more binding data sharing requirements (4.5). For each result section, we distinguish between the interviews with representatives of German funders and interviews with research data experts from other organisations, which we refer to as "stakeholder organisations" throughout the article. We highlight noteworthy aspects and illustrate them with passages from the interviews. Finally, we discuss the particularities of German funders in promoting data sharing based on a comparison with international funders and point to some ethical and legal implications of the results relevant for funders' development of data sharing policies (Chapter 5).

## 2. Background: German funders and data sharing policies

To allow for a better understanding of our results, we provide some background of the German research landscape as far as it is relevant to data sharing and funders' data sharing policies. This overview is based on our research on the German science system and findings from our interviews.

### 2.1 Pillars of the German research funding landscape

As a federal republic, Germany has a large number of public funding agencies both on federal ("Bund") and state level ("Länder"), such as federal and state ministries. Furthermore, there are numerous charities that provide private research funding. Another characteristic of the German funding landscape is the existence of project execution organisations ("Projektträger"), which are service providers that frequently advise public and private funders in designing grant programs, administer review procedures of grant applications and monitor funded projects.

Project grants play a major and increasing role in the financing of research in Germany. In terms of awarding project grants (project funding), two public agencies stand out in the German funding landscape. First, there is the Federal Ministry of Education and Research (Bundesministerium für Bildung und Forschung, BMBF), which funds research across all disciplines but also invests in the development of research data competencies and infrastructures. Second, there is the German Research Foundation (Deutsche Forschungsgemeinschaft, DFG), which is the largest non-governmental funder of research projects in Germany, with a current annual budget of 3.6 billion Euros [41, 54]. While the DFG is jointly financed by federal and state governments, it is also the central self-governing organisation of science in Germany and funds all different kinds of research.

Both the DFG and BMBF play a key role in developing the publicly funded German National Research Data Infrastructure (Nationale Forschungsdateninfrastruktur, NFDI), as recommended in a position paper published by the German Council for Scientific Information Infrastructure [55]. With a funding volume of up to 85 million euros per year, the NFDI aims to fund, develop and coordinate infrastructure platforms for systematic management, storage and accessibility of research data (based on the FAIR principles) and provide services for scientific communities [56, 57]. The NFDI is also part of the European Open Science Cloud (EOSC) [58].

### 2.2 Data sharing strategies, documents and policies in Germany

As written in its coalition agreement in 2021, the new German federal government [59] acknowledges the importance of research data, which indicates a growing political awareness regarding research data in Germany in recent years (and compared to previous governments). There also appears to be an increasing tendency at the level of both research societies and universities to develop policies for open science in general and for research data in particular

[42, 60–62]. However, out of sixteen funders we investigated, only three of them had some kinds of data sharing policies on their websites (The BMBF, the DFG, and the Volkswagen Foundation).

The BMBF's "Action Plan on Research Data" ("Aktionsplan Forschungsdaten") [63] aims to promote a "culture of data sharing", inform and develop more data competencies for researchers and establish infrastructures like the NFDI. According to this general guideline, the BMBF asks grantees to commit to the FAIR criteria and strives for more recognition of FAIR data publications as part of grant evaluations. In its complementary funding programs and calls for funding, the BMBF increasingly–but not universally–requires Data Management Plans (DMPs) and compliance with FAIR criteria. Beyond these specific programs, however, the BMBF has yet to include a general data sharing policy in its funding conditions.

According to our interviews, a key document of the German research data landscape and the German science system overall is the DFG's Code of Conduct "Safeguarding good scientific practice", which was last updated in 2022 [64]. Even though the Code is rather a general guideline on good scientific practice than a specific document on data sharing, it explicitly addresses data sharing for reasons of reproducibility and secondary use, describes general standards for good research data practices and proposes adherence to the FAIR principles. Researchers and research organisations in Germany need to comply with the Code in order to be eligible for receiving DFG funding. In addition to its Code of Conduct, the DFG provides a guideline on research data, supplemented by recommendations from and for different scientific disciplines [65]. Part of this guideline is a mandatory checklist for funded projects. This checklist is comparable to the Data Management Plans (DMPs) of international funders and includes questions on the management and sharing of research data.

In total, prior to our interviews, we only identified three funding agencies with documents and guidelines that qualify as data sharing policies that match our definition. For the BMBF and the DFG, we consider the combinations of their general guidelines and their more specific instructions to be data sharing policies in a broad and inclusive sense of the term. However, these policies do not appear to be particularly advanced compared to those of leading international funders [39, 42], which tend to have explicit and focussed data sharing policy documents on their websites.

## 3. Methods

### 3.1 Sample selection

In order to obtain a balanced sample for our interviews with data sharing experts, our sampling process followed four criteria. First, we aimed to include the most important public German funding agencies, both in terms of annual funding and political significance, which are the DFG, the BMBF, and other federal ministries in charge of research funding. Second, we contacted some of the more important private German funding agencies, usually charities, to acknowledge different data sharing policy approaches between public and private funders. Here we turned to funders for health research, as data sharing seems to be comparatively advanced in this area [39]. Third, we considered the role and function of so-called "Projektträger" (project execution organisations) in Germany, as they frequently advise funders in designing grant programs, administer review procedures and monitor funded projects.

Lastly, we included some interviews with data sharing experts from what we call stakeholder organisations, such as research societies (e.g., the Max-Planck-Society or the Helmholtz Association), universities, research institutes, scientific advisory councils and other organisations involved in the topic of data sharing. The inclusion of both funding agencies and other involved stakeholders allows us to understand and compare both internal and external

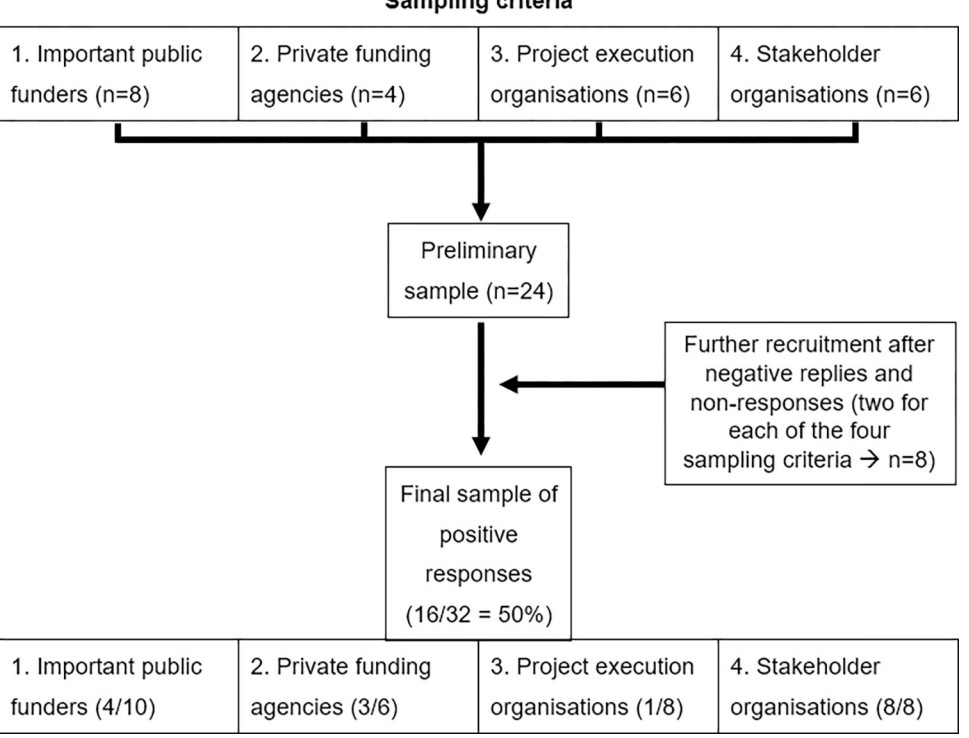

**Fig 1. Sampling procedure.**

perspectives on the role of funding agencies. In order to achieve an approximate number of 15 interviews, the initial sample comprised eight public funders, four private funders, six project execution organisations and six stakeholder organisations. After a few unsuccessful recruitment attempts, we contacted additional organisations (two per sample criterion) to obtain a serviceable sample size (Fig 1). We contacted two randomly selected science ministries on the state level ("Landesministerien") to recruit additional public funders.

The response rates to our interview requests show strong variations. Compared to a serviceable response rate from public funders (4 positive responses out of 10) and private funders (3 positive responses out of 6 requests) as well as the stakeholder organisations (8 positive responses out of 8 requests), our response rate from project execution organisations is notably low (1 out of 8 requests), so that we decided to include this interview in the funders' interviews for reasons of safeguarding anonymity.

## 3.2 Data collection

The interviews took place between 15 September 2021 and 1 April 2022 and lasted on average approximately 51 minutes. All interviews were conducted via video conference tools like Zoom or WebEx, depending on the preferences of the interviewees. In order to perform a comparative content analysis at a later stage, the interviews followed structured interview guidelines (S1 and S2 Appendices). We constructed this guideline based on a review of scientific literature on data sharing, an initial exploration of funding agencies' policy documents and discussions within our interdisciplinary project. In contrast to many leading international funding agencies [39], most funding agencies we approached did not have explicit contact persons for requests pertaining to open science or data sharing. We therefore requested to talk to

experts on the topic of data sharing, open science, research data or, if none of these were available, funding policies in general. Fourteen of the interviews were one-on-one conversations and two interviews involved two participants. We jointly translated passages from interviews for the purpose of this article, as all interviews were conducted in German. After each interview, the audio files recorded with the software OBS [66] were encrypted, anonymized and transcribed, using the software MAXQDA 2020 [67] and the transcription rules defined in our transcription and coding guideline (S3 Appendix). All interviews were transcribed by student assistants and double-checked by the authors.

### 3.3 Data preparation and analysis

We used a specific method of qualitative content analysis to code the interview materials [68]. First, we conducted an initial engagement with the data and created a coding guideline to define a set of rules for the coding process (S3 Appendix). We drew on our project's general research questions, our interview guideline and supplemental discussions to deductively develop preliminary main categories. We defined and exemplified these categories in the codebook and drafted a first version of the category system. Afterwards, we conducted an initial round of coding using the main categories and MAXQDA 2020 to check the adequacy of these categories and identify patterns and peculiarities within the interview material. This led us to develop relevant subcategories from the empirical material in a mostly inductive manner, culminating in a final category system and codebook (S4 Appendix). We then performed a second round of coding using the category system and the codebook. All coding was performed by a postdoctoral research associate and a research assistant, who jointly reanalysed and compared all codings to reach consensus [69]. In a final step of content analysis, we interpreted our data from a sociological and ethical perspective and arranged our findings in five result sections to answer the research question of this publication.

### 3.4 Ethics statement

We obtained informed consent for every interview and all interviewees received detailed written information on the study several days prior to the interview. Before starting the interviews, we offered additional explanations and the opportunity to ask questions to each interviewee. Data processing is officially registered with the data protection office of the German Cancer Research Center (registration no. D120-P989), including the use of oral consent for the interviews. Oral consent was chosen over written consent for three reasons: First, we conducted all interviews via videocall. Second, we reduced the amount of additional sensitive data by having both the consent and the audio data in one encrypted file. Third, all interviewees explicitly accepted oral consent, as it also reduced bureaucracy for them.

The ethics committee of Heidelberg University Hospital waived formal approval in the form of an ethics statement because we do not conduct medical research and our study neither includes medical professionals nor patients or any other vulnerable groups. We processed all data in accordance with the European General Data Protection Regulation (GDPR) and followed the ethical standards of the DFG Code of Conduct [64], which is the essential guideline for research in Germany.

## 4. Results

We present our findings in four thematic sections. In each, we first report the perspectives from funding agencies and then the perspectives of stakeholder organisations. Please note that our results mention organisations that we have not necessarily talked to as part of our investigation. All interviews were conducted in German and translated by the authors.

## 4.1 Perspectives on the general status quo of German funders' data sharing policies

**4.1.1 Interviews with German funding agencies.** The first result section describes how the interviewees perceived the status quo of data sharing policies in Germany. Some funder employees expressed hope that recent developments, such as the German government's greater appreciation of research data and the development of the NFDI, will motivate research data management and harmonise research data standards.

> *"I believe that with the new government and the agreements in the coalition agreement, there will probably be some changes in the area of health data. [. . .] With the very different and uncoordinated procedures and regulations in the different federal states, the lack of clarity between the tasks of the federal government and the tasks of the state governments, we really have a lot of issues to address at the same time and we need to harmonise our efforts."*

Although only a minority of funding agencies in Germany seems to have policies on data sharing, all funder employees we interviewed confirmed that their agencies are either developing some kind of data sharing policy or are at least strongly considering doing so in the future.

> *"[. . .] we as a foundation have not yet implemented any data sharing policies within our funding guidelines. However, we have been observing this development for some time and are also following the discussion about it, and in fact we are currently planning to strategically include a few points in the application process with regard to this topic."*

> *"The topic of Open Science is constantly in the media or the topic comes up in our work, but we are not very far yet. [. . .] As far as pure research funding is concerned, we have not yet set up a project where we want to concretise the funding guidelines. But we would like to do something like that, if we have suitable instruments."*

In terms of common standards for data sharing policies, several interviewees pointed to DFG Code of Conduct or the FAIR principles as important references for their organisations' policy development. Furthermore, a few funders perceive data sharing to be a part of good scientific practice.

> *"[. . .] there is no real research data policy or open science policy yet. So far, we have been guided by overarching principles, such as the DFG code on good scientific practice. We are committed to the FAIR principles, but this has not yet become part of a governance process. [. . .] And something like a policy process will certainly be integrated into this, but it still needs to be adopted."*

> *"In our research contracts, we refer to the generally accepted scientific standards and this covers everything for us that is related to good scientific practice and, if this is part of this practice, and I think it is, then of course a certain amount of open data, open science etc. would also be part of it."*

Our interviews show a pattern in which smaller funders frequently follow larger agencies, with the DFG being the most important point of reference. However, there often seems to be little coordination and cooperation between different funders.

*"As a rather smaller research funder, however, we do indeed orient ourselves towards larger organisations, especially the DFG [. . .] simply because we naturally tend to stick to the larger players and what they demand from researchers."*

*"I don't think we formally coordinate anything, but of course you look at what other funders are doing and what their clauses are, that's quite common. You can only learn from what others have done [. . .] as a research funder you have to make sure that not everything is completely contradictory, right? For example, for us it would also be important to see how the DFG handles this."*

Despite this lack of formal coordination, funders are striving towards shared goals such as setting up a public research data ecosystem and infrastructures to allow for better data sharing, especially through the NFDI.

*"I think it is quite elementary that we manage to set up a, how shall we put it, public ecosystem for research across all stages of the research lifecycle. [. . .] And NFDI is a part of it, the European Open Science Cloud is a part of it, the infrastructure that is connected to it with Open Research Europe is also a part of it, and we have to put more pieces of the puzzle together."*

Overall, there are diverging opinions on the efficiency and success of funders' past efforts and the current status quo of data sharing policies. While a few interviewees perceived German funders to lag behind, there are also more favourable perceptions.

*"Well, internally we have long been of the opinion, and of course we have always liked to make comparisons with international funding organisations, that there is an urgent need for action here and that any inaction on the part of [funder], and I am exaggerating now, also has consequences [. . .] we are finally closing the gap, so to speak. But we are aware that we are far behind, yes."*

*"I have the impression that in the last two years especially the BMBF recognised and reacted to data sharing with increased intensity [and] drafted funding programs [. . .] in which data sharing was formulated explicitly as an important component, so I think they are responding well to this issue. Whether that will be sufficient, time will tell."*

**4.1.2 Interviews with German stakeholder organisations.** Throughout our investigation, the interviewees from stakeholder organisations had rather heterogenous perspectives on the German data sharing landscape's status quo. A few interviewees praised the efforts by the European Commission and its Horizon 2020 and Horizon Europe programs as critical inspiration and model for the German research data landscape.

*"The European research funders' call to have data management plans for projects that we fund was beneficial. It triggered a whole slew of efforts in Germany as well and put the question »What do you actually do with the data?« on another level. Horizon 2020 has simply had a beneficial effect."*

Some interviewees rated the efforts and data sharing policies of German funders to be positive and described funders as major drivers with fruitful approaches.

*"[. . .] the European Commission, but also the DFG and partly also the BMBF have issued guidelines for research data that say research data should be published. And I think that was*

*an important trigger for this whole research data management and for the open data movement."*

*"My observation is that in practice the progress that data sharing has made in the last ten years has mainly come from the requirements of the research funders. That this has at least catalysed this basic attitude and in many cases led to researchers to look at this topic in the first place."*

Several interviewees described the DFG as the leading German funding agency and perceived it in a more positive light than the BMBF. However, some interviews also included criticism of the two major funders.

*"I would of course have liked to have seen more from the DFG. But with the revision of the DFG Code on good scientific practice, which until a few years ago was the only point of reference that stated how long data should be kept [. . .], the DFG has once again considerably supported this process [. . .] The BMBF has not yet taken on this role [. . .] It could still take a page out of the DFG's book, especially with regard to [. . .] requirements for data availability. . . it should also get more involved in the European context."*

## 4.2 Perspectives on the role of funders in promoting data sharing

**4.2.1 Interviews with German funding agencies.** One of the main topics within our interviews was the role that funding agencies can play in promoting data sharing. Despite spending public money, public funders usually did not perceive governmental strategies or public interest to be the main drivers for improving research data accessibility. Instead, they frequently named expectations and pressure from scientific disciplines and research data infrastructure institutions as reasons for developing shared standards and framework conditions.

*"There was strong pressure from the scientific community itself, especially from the data-heavy areas that need quality-assured research data. It was as if they wanted to say: »You are also there for us, so please create the conditions for our work«, and of course also from the infrastructural institutions that say: »If you want to invest money so that we build infrastructures and support capacities here, you also have to commit to it.« So you could feel this kind of pressure. Not really from the political side so far. Not even in the sense of: »It's public money, do something about it«."*

Some interviewees emphasised that funders are engaging in discourse with different scientific communities. As facilitators, they help to provide infrastructures (e.g., via the NFDI), raise awareness of data sharing, help to develop community standards and advocate for more data competencies.

*"I believe that our essential role is to create the infrastructures [. . .] and engage in the discourse. That is, the discourse about the topic, bringing together of important actors in this field to help the different communities to really work out a guideline [. . .] which can then somehow be a valid policy, so to speak, for the community. [. . .] We often take over where no professional society has been active in the field. We fill the gaps, so to speak."*

*"There are areas that are now very data-rich, where it's simply been common practice for a long time to work with large amounts of data [. . .] but there are still many areas that are rather data-averse, that have their data, preserve it according to their usual standards, but*

*don't use the full potential of this data. And that's exactly where we want to get to: promoting and advancing data competencies across the board."*

Furthermore, large funders want to go ahead with other agencies in a joint effort to advance a cultural change towards more data being shared. In order to change norms and standards around research data, our interviewees perceived the need for funders to address researchers and scientific communities, but also grant reviewers and other employees of funders. Some interviewees considered their agency's role as to accompany, moderate or even push this cultural change.

*"We want to create another clear explicit regulation that should go exactly in this direction. And you can imagine that something like this would not only create a kind of cultural change among the researchers, but it would also create a kind of cultural change within the funding agency."*

The interviews also raised the question whether the efforts of funding agencies alone can stimulate this cultural change towards more data sharing. While some interviewees agreed that funders should play a role in this development, they also stress that funders cannot implement cultural change by themselves, but also rely on scientific communities to adapt norms and standards.

*"When it comes to cultural change, I would actually see it first and foremost as an issue inherent to science. I mean, you cannot flip a switch and change the culture of science from a ministry somehow."*

*"[Data sharing] must become a criterion in applications, for example. And that is not something that [we as a funder] can implement alone, this must become part of the standards of the scientific community."*

Smaller, particularly private funders usually described their role rather cautiously in our interviews. They often referred to the leading role of the DFG and the BMBF and perceived their own function to be more of a partner of funded researchers and engage with them in a relationship built on trust.

*"We see ourselves as supporters, as providers of funding, so that these people can carry out their projects successfully. As I said, we basically have the attitude that we have confidence in the researchers, that they will act responsibly and correctly in the sense of good scientific practice."*

*"I'll start with the most important thing: funding is also trust. If you control everything, you lose the trust of the applicants and the scientists."*

**4.2.2 Interviews with German stakeholder organisations.** A number of interviewees from stakeholder organisations acknowledged that funding agencies can play an important role in promoting data sharing, for example by supporting the development of research data infrastructures, raising awareness for data sharing through grant calls and supporting data management efforts.

*"[. . .] of course, research funding organisations are a big driver in providing the appropriate infrastructures and in dealing with research data management. And we very much welcome*

*the fact that this is anchored in more and more calls, that data management plans are required. But also that funding organisations support research data management efforts."*

Some interviewees assumed that a cultural change towards data sharing requires some amount of outside intervention from funding agencies and a few even suggested some kind of pressure from funding agencies.

*"Of course, it's all about reputation in the end, about evaluation. [. . .] So that has to come from the outside; researchers can't regulate that themselves. Of course, reliable, sustainable infrastructures must also come from outside. [. . .] And this must also be secured from the outside; the funders are then also obligated to do something."*

*"I think this has to be done on many levels, both top-down and bottom-up in the organisations themselves, but it also has to be accompanied by. . . I don't want to say coercion, but a certain pressure from outside. I prefer the funders to the legislators and I believe that in an international context this could lead somewhere."*

Our interviews indicate that several interviewees consider research funding agencies to have a moderating function between science, politics and legal regulations. Therefore, funders need to clarify legal frameworks and define the conditions for data sharing.

*"The funding agencies have to answer these questions first: If the funding is linked to a data transfer, then it must be clear which data is received by whom and under which conditions? [. . .] This is all embedded in a legal framework and [. . .] we must first clarify these framework conditions very precisely. [. . .] Because the funding agency also has to know what it then demands from the researchers, under which framework conditions this is possible. [. . .]. So, we are in a difficult situation and if the funding agencies implement it in guidelines, the framework conditions must be clear.*

Lastly, a number of interviewees stated that funders' role in fostering data sharing mostly depends on their size. Whereas small funders like charities only have limited leverage, large funders like the DFG, the BMBF or the European Commission are perceived as leaders with considerable influence.

*"From my point of view, it also depends on how important these funding bodies are. So, if you get the big players to do it, then I think that definitely has an influence. [. . .] I mean, the EU does this in Horizon Europe and it's already very much part of it and I do believe that it definitely has an effect."*

## 4.3 Perspectives on funders' current and potential measures to promote data sharing

**4.3.1 Interviews with German funding agencies.**   Our interviews showed a broad consensus that funding agencies have several instruments to promote data sharing. For example, funders can provide financial incentives via research grants.

*"We have the opportunity, through financial incentives, through our funding, to influence certain behaviours, in this case, improved sharing or availability of data. And I see an important task for [funding agency] [. . .] to push it a little bit, to accompany it, and perhaps also to moderate it a little bit at the same time."*

*"The incentive we have, of course, is the subject of money. [. . .] we provide money for certain projects and we can also demand certain conditions with this money. And that is exactly what we do. We want to create an incentive to share more data."*

However, many statements suggested that funders could and should do more. Most employees of funding agencies agreed that there is a lack of incentives for data sharing and that researchers need more and better incentives to share their data. For example, there was a broad agreement that data sharing deserves more acknowledgment in funders' assessment of researchers. Our interviews indicated that several funders strongly consider introducing new research assessment metrics to provide more incentives for data sharing.

*"We need another currency, so to speak, namely the willingness to share data, and sharing data has to benefit the scientific reputation of the researchers who are sharing. So that in an appointment procedure [. . .] the willingness to share data can be documented, can become part of the evaluation."*

*"The idea is that it then plays a stronger role from the beginning, in the application process and in all following steps [. . .] in the evaluation of proposals, we also look at other aspects than just the impact factor and so on."*

A few funders intend to move beyond creating incentives and consider using their influence to push data sharing by implementing more concrete expectations or requirements regarding research data as part of their funding policies. This includes mandatory Data Management Plans or expectations for research data to be FAIR.

*"It would be important to me to have a strong default setting that says data should be FAIR. That we don't leave it entirely up to the researchers but we as a research funder communicate that »It's important to us« and say »we're not forcing you«, which is always difficult politically as some people always refer to academic freedom [. . .] Instead we say: »This is actually the standard that we expect«"*

Our interviews suggested that measures like these neither seem common nor systematic at this point. Some funders ask grantees to submit a Data Management Plan but do not formulate explicit expectations or demands for actual data sharing.

*"So, what we are doing now in terms of obligations is that we have a mandatory reflection on [data sharing] in the projects. The step that we are not actually taking yet is the step that I think you are talking about. So, we will still do not have an obligation to publish the data somewhere, so to speak."*

Funders also hardly monitor whether grantees comply with funders' requirements or adhere to established research data standards and rather rely more on trust than monitoring. However, project execution organisations could perform checks if there were concrete requirements regarding data sharing.

*"What the project execution organisations do at any rate is to check: Have all the formal criteria been met, have all the required documents been submitted? And the first step would actually be: is there a research data management plan? That's a relatively formal criterion that can be checked quickly, and it will be checked regardless, and I don't know exactly to what extent the quality of such plans will now be improved."*

When asked about (hypothetical) sanctions for non-compliance with funders' data sharing policies, our interviewees mostly excluded this option or deemed it merely theoretical, be it because of the overall lack of monitoring or because they deem this instrument inappropriate for different reasons.

*"It is difficult to apply thumbscrews because we do not have the possibility to check in detail whether our grant contracts are being adhered to [. . .] If certain requirements were not met, one could think about reclaiming funds or whatever. We have never done that; somehow it has always worked out."*

**4.3.2 Interviews with German stakeholder organisations.** Similar to the funder interviews, most interviewees from stakeholder organisations highlighted the importance of monetary incentives provided by funders, be it in the form of additional money for data management or research funding itself.

*"And it is also the case that, at least with the DFG, I know that various costs incurred for research data management are reimbursed. And I think that would be a very important incentive, that financial support is also provided."*

*"But as usual, money is still the best argument, and of course research funding organisations are a big driver for providing the appropriate funds, infrastructures and for dealing with research data management."*

The interviewees broadly agreed that funders can provide important incentives for data sharing by moving away from research assessment criteria like the number of publications and towards more recognition of data sharing and other Open Science contributions.

*"One incentive could be that you try an Open Science Record. . . or instead of the Publication Record, the Data Record is also an important feature of one's scientific curriculum vitae. And if you say somewhere that sharing data will be recognised, that it will perhaps also bring you advantages in grant applications because it shows additional scientific achievements alongside the publications, that could definitely be an incentive."*

While the majority of interviewees agreed that researchers need more incentives to share their data and funders can play a role in this, some interviewees also highlighted the need to increase intrinsic motivation and provide concrete general and individual benefits of data sharing.

*"The researchers have to be intrinsically motivated. If a researcher finds something meaningful [. . .] he will work on it day and night, but he has to see the meaning and not just think »I need this for the application«."*

*"I believe that the best incentive is a better research practice. In other words, the experience that my discipline works better, that I can work better, that others can work better if we share data and make data available."*

Our interviewees' evaluations of funders' measures to promote data sharing were very heterogeneous. Some interviewees praised funders for successfully promoting data sharing and encouraged funders to become more binding in their demands or highlighted funder and journal policies as major motivators for data sharing.

*"My impression is that the measures of the funders so far demonstrate good judgment and I also see that data sharing is being established. Having said that, I think the requirements could now become a little more binding."*

*"But I have also seen in my consulting practice that one of the main reasons for Open Science for many researchers is based on the requirements of the research funders and partly also the requirements of the journals."*

Others expressed more critical views on funders' measures, lamenting that some requirements are not useful for research or even discourage researchers. If funders ask researchers to explain how they intend to share their data via data management plans, these plans should be homogenous and benefit researchers.

*"There are templates where you have the feeling that it's all about the requirements that come from the funding agency and not about the usefulness for the research. [. . .] And if I tell a researcher »You need this because the DFG wants it«, then he will do it unwillingly [. . .] It looks beautiful in the application, but there has been no reflection behind it and it has no effect."*

*"In principle, everyone has their own template [. . .] and everyone asks different questions, which is also logical. The DFG certainly has different needs than an EU project, but of course it's not easier for us or for the researchers if you have to adjust to something new with every funding program, with every funding agency."*

### 4.4 Perspectives on general barriers and limitations for funders

**4.4.1 Interviews with German funders.** Our interviews show that funders face several general hurdles, such as the overall complexity of data sharing. As the field is highly dynamic and there are different community standards to consider, implementing data sharing policies is a particular issue for smaller agencies due to limited capacities.

*"The field is highly dynamic, a lot is happening and for us as a small funder it is not very easy to keep up [. . .] with different ethical, legal and also technical and institutional framework conditions [. . .] what we are trying to develop must somehow work for all areas, which means we have to make it soft enough so that everyone fits in, while at the same time keeping diversity in mind."*

Larger agencies might have more resources at their disposal, but they also perceived their funding policies to be complex and the implementation of changes or new data sharing policies as complicated and time-consuming.

*"And now [department] comes along and says: »Please include in the grants that you should also share your data«, which of course leads to funding per se becoming more and more complex. [. . .] It can have major consequences if something like this is written into a funding decision and you have to think about it."*

*"So, this commitment exists, it must be said, of course, it is also a question of implementation and changing the funding regulations of the [funding agency]. This is not done with a snap of the fingers; these are also processes that require coordination and are therefore designed for the longer term."*

Funding agencies are also not homogenous actors, but often include several persons or departments with sometimes conflicting perspectives on data sharing and policies. This makes it harder for funders to implement new policies or update existing ones.

*"It is interesting to observe that the discussion in [committee] is very, very much in favour of defending the existing practice. [. . .] We did not succeed in convincing [committee] that this was important at the first go. [. . .] changes at this point require a great deal of tact. [. . .] We will now discuss this a little further with [committee], because I think it is absolutely clear that this notion about the ten most important publications is no longer up to date."*

**4.4.2 Interviews with German stakeholder organisations.** Especially the part of motivating researchers and incentivising them to share their data remains a challenge for funding agencies in Germany. As showcasing its concrete benefits to researchers is difficult, incentives alone might not be enough to motivate data sharing.

*"The direct benefit of sharing one's data is very, very difficult for someone to grasp. It is very difficult to get people to make the effort for a hypothetical benefit in the future. [. . .] That's why I think it will take a very long time or not work at all with just appeals and incentives."*

While many researchers are perceived to be in favour of data sharing, the competitive context and the additional workload for researchers may pose hurdles for funders to encourage data sharing.

*"I think that if you were to interview scientists now, they would actually all commit that of course they want to make their scientific results accessible, but you would often find in practice that this is not consistently implemented because of the competitive situation in which they find themselves. This is one major reason. And the second main reason is, of course, that if they want to make information accessible to third parties, it has to be prepared and that can be very labour-intensive and the manpower is often not available."*

Finally, a few interviewees noted that it is not just about the appreciation of data sharing and the competencies held by researchers, but also about reviewers' abilities to adapt to new requirements.

*"Another thing is [. . .] to consider the distribution of competencies among the reviewers in the evaluation process. It's not only pure expert scientists, some of whom have acquired their scientific reputation and have come to a position like this in a much older system and who are basically totally incapable of really even understanding the digital requirements, if you'll pardon my wording. . ."*

## 4.5 Perspectives on implementing more binding data sharing requirements

**4.5.1 Interviews with German funders.** Beyond general hurdles for data sharing, our interviews suggested that a major challenge for German funders concerns the development of more concrete and binding requirements regarding data sharing. Since data sharing requirements can only work if researchers have the appropriate abilities to fulfil them, a few interviewees were concerned about researchers' lack of skills in handling and sharing research data. A few funders also stressed that more binding demands depend on the development of adequate community standards.

*"All the demands we make, like »You have to handle the data better, you have to make more of it', can only work if everyone involved in the process have the appropriate competencies; anything else makes no sense. And I think it's a big task for the funding agency to ensure that, especially in areas where these competencies are perhaps not yet available. [. . .] And that's exactly where we want to get to, promoting and advancing data competencies across the board."*

*"The question is always how much can you demand from scientific communities. [. . .] You just gave the example of data management plans, so one could also require these, or data to be put in certain repositories. So you could make it all much more concrete. At this point, we believe that it is important to give science the necessary freedom to develop and determine for itself what is best."*

As a precondition for implementing more binding data sharing policies, many interviewees pointed to the availability of data sharing infrastructures as an important precondition for data sharing. One interviewee also highlighted the need for funders to be able to monitor researchers' actual compliance with potential data sharing requirements as a further prerequisite.

*"An obligation without being able to monitor it somehow and an obligation without having the infrastructures, which we still don't have in parts, [. . .] this is something that is not compatible. And in our exchanges with international funders they always say: »Yes, we have this obligation«, and when you then ask: »So what do you do with it or how do you monitor it or how do you support people in finding the right place for their data?«, it becomes a bit blurry. To demand things that we cannot implement and support ourself is not something we aspire to."*

Some interviewees expressed concern about the additional bureaucratic burdens that are associated with monitoring data sharing and that additional expenses for it could come at the cost of other research funding, even if monitoring could be delegated to project execution organisations.

*"I am convinced that we should try to set up the system in a way that monitoring is not really necessary, because it is not feasible. I believe that you can try to maximise control, but only at the cost of making documentation exorbitantly large, in more plain words: by building up bureaucracy."*

*"One consequence would be to make more and more use of corresponding project execution organisations, which have the task of monitoring this and are paid to do so, or to make fewer calls for proposals. Probably both, because the money I spend on a project execution organisation I no longer have for research funding."*

A strong pattern within our interviews shows funding agencies also perceive their own scope of action to be limited by academic freedom ("Wissenschaftsfreiheit"), which is enshrined in the German constitution as part of the freedom of expression, arts and sciences [70].

*"We have academic freedom, so it is not the task of the state to change the culture of science. But I also think that there are points in which science can start to change the culture and where a ministry can possibly offer support. [. . .] So you can do that, but I think this cultural change has to come from the scientific community itself."*

All of the funder employees we interviewed highlighted the importance of academic freedom as a major concern for their agency and a limitation to making data sharing policies more binding. However, many interviewees also expressed differentiated views and stated that one needs to put these concerns for academic freedom in context.

*"What can we expect from applicants without coming into conflict with the area of academic freedom? This was already an important concern for us, to somehow illuminate this area. Let's say, we would certainly have to look at this in more depth, but I think the culture is changing a bit."*

*"So, we encourage it, we appreciate it when data is shared, but we don't force anyone to do it. The final decision on whether or not to share data is up to the researcher, so I don't see any conflict with academic freedom here. Of course, should we go a step further and eventually come to a point [. . .] that we really make it an obligatory requirement, [. . .] then we could discuss again whether this is an interference with academic freedom."*

However, a few interviewees also stressed funders' responsibility to safeguard good scientific practice and pointed out that the right to academic freedom does not warrant a right to bad scientific practice, so that, if data sharing is a part of good scientific practice and the modalities around it are clarified, funders may (and ought to) make it a condition of their funding.

*"The rights of scientists come with obligations, namely to make the results of research available for public discussion. [. . .] especially in data-driven disciplines, you cannot evaluate or review research without the data, so just keeping the data for yourself is not possible. If this were to be incorporated into the understanding of academic freedom in this way, then we would create a right to bad scientific practice and that is something we do not want."*

*"It is our responsibility [. . .] to make it a precondition of our funding that good scientific practice is adhered to and, if [data sharing] is part of that, if there is a general consensus about it and if the modalities are also clarified, then to ensure that data are published and made available."*

**4.5.2 Interviews with German stakeholder organisations.** Safeguarding academic freedom is not just a priority for funders, but also for stakeholder organisations. Some interviewees perceived it as clear limitation for advancing aspects of Open Science such as data sharing.

*"Of course, [organisation] advocates Open Science, but under certain conditions, so not somehow naively thinking: we'll make everything open and then that's fine. The most important thing is to ensure academic freedom and quality."*

*"Of course, there is an area of tension. Academic freedom is enshrined in the constitution. It's a great good; it's not something that's up for grabs in any way. The question is to what extent one can realise the goals of Open Science within the framework of this freedom of science."*

However, some interviewees stressed funders' right to require data sharing as part of their funding policies and did not perceive this to be a violation of grantees' academic freedom. A few interviewees even encouraged funders explicitly to be more binding with their policies.

*"I think the freedom of research is an important good, I think that research funders should not dictate to the researchers exactly how they have to do their research [. . .]. But I do think that*

*if money is given for it, whether by society or the state or whoever, one can demand that the corresponding results, even if they may have failed, must be published, if necessary, so that they are also public. And I don't think that this contradicts academic freedom, as long as things like embargo periods etc. are offered."*

*"My impression is that one can, with a clear conscience, place a little more emphasis on [data sharing]. The DFG, for example, currently formulates: 'Research results from funded projects should be made publicly accessible', so you could make it an 'ought', maybe even a 'must' in a few years…"*

Furthermore, a few interviewees argued that if data sharing can be seen as a part of good scientific practice, requirements to share data do not necessarily violate academic freedom. Accordingly, they point out that the role of data sharing in good scientific practice needs to be clarified.

*"Good scientific practice is […] a very strong guideline and the scientific disciplines must clarify their position on Open Science. Is it good manners, good scientific practice, to publish data? And if so, to what extent? […] How far is it expected and how far is it rewarded by the scientific community?"*

*"Does good scientific practice expect open science and if so, to what extent? I think this is the main point at which you can combine both the desire for openness and academic freedom, because this freedom does not go so far as to allow us to disregard good scientific practice."*

**4.5.3 Overview of the interview results.**   Before moving to the discussion of our most important findings, we provide a brief overview of our interview results (Table 1).

## 5. Discussion

The aim of this expert interview study was to investigate how German data sharing experts within and outside funding agencies perceive and evaluate German funders' data sharing policies as well as their overall efforts to promote data sharing. In the following, we discuss our main findings with a view of the relevant literature and the specifics of the German funding and data sharing landscape, and draw some comparisons to international funding agencies. In doing so, we highlight several ethical, legal and science policy implications of our findings for the future development of funders' data sharing policies. Since there are numerous challenges surrounding data sharing and the design of data sharing policies, we also point to potential solutions mentioned in our interviews.

Regarding the status quo of German data sharing policies, our interviews suggest that while several funders are considering introducing data sharing policies, only a few funders have some sort of data sharing policy in place. Leading German funding agencies like the DFG or the BMBF already have data sharing policies in a broad sense of the term, but they are not yet as explicit as the ones of leading international funders, which tend to have more comprehensive data sharing policies. These observations support other reports on different stages of policy development among funding agencies [35, 39, 42–43, 71, 72] and are particularly noteworthy for two reasons. First, our interviews indicate that smaller agencies orient themselves on the policy approaches of large funders like the BMBF and especially the DFG with its Code of Conduct. Second, other studies suggest that data sharing policies are important guidelines and motivators for researchers [6, 8, 30, 32–37, 73, 74]. Our interviews suggested several reasons for the lack of policies and hesitance towards the development of policies. German

**Table 1. Overview of interview results.**

| Section | Funder interviews | Stakeholder interviews |
|---|---|---|
| **4.1 Status quo of data sharing policies** | Funders are either developing some kind of data sharing policy or are at least strongly considering doing so in the future. <br>FAIR principles or the DFG Code influence policy development. <br>Smaller funders follow the lead of larger funders, especially the DFG. <br>Mixed evaluations of the status quo of data sharing policies. | Rather heterogenous perspectives on the status quo. <br>Efforts by the European Commission are very important. <br>The DFG is the leading funder with regard to data sharing policies. |
| **4.2 The role of funders in promoting data sharing** | Expectations from scientific disciplines rather than political pressure are a main driver for the development of data sharing policies. <br>Funders engage in discourse with scientific communities, facilitate and promote data sharing by raising awareness, helping to develop community standards and advocating for more data competencies. <br>Large funders want to promote a cultural change towards more data sharing. Funders can accompany, moderate or push cultural change. <br>Funders rely on scientific communities to promote cultural change. <br>Smaller funders usually follow the DFG and the BMBF while perceiving their own function to be more of a partner of grantees. | Funders can play an important role through funding research data infrastructures, raising awareness, and supporting data management. <br>Funders could do more to promote a cultural change towards data sharing. <br>Funders can moderate between science, politics and legal regulations and clarify framework conditions. <br>Funders' role in fostering data sharing depends on their size. |
| **4.3 Funders' current and potential measures to promote data sharing** | Funders could and should do more, e.g., providing more and better incentives for researchers to share data. <br>Funders are aware of the lack of incentives. Most of them consider introducing evaluation metrics that acknowledge data sharing. <br>Implementing concrete expectations like DMPs or FAIR principles. <br>Even advanced funders lack explicit expectations or requirements as part of their data sharing policies or grant conditions. <br>Most German funders rely more on trust than monitoring and hardly monitor whether grantees adhere to established standards. <br>Sanctions for non-compliance are not an issue due to a lack of monitoring and because they are considered to be inappropriate. | Financial incentives provided by funders are important. <br>Evaluation criteria that recognize data sharing are another important incentive. <br>Funders' requirements need to be consistent and clarify the benefits of data sharing. <br>Funders could be more demanding about data sharing. <br>Funders should address the intrinsic motivation for data sharing. |
| **4.4. General barriers and limitations for funders** | The complexity of implementing data sharing policies seems particularly challenging for small agencies. <br>Larger agencies have more complex funding policies and internal decision-making processes. <br>Funding agencies include several departments with sometimes conflicting perspectives on data sharing and policies. | Incentives are a challenge and might not be enough to motivate sharing. <br>Competition and additional workload are hurdles for data sharing. <br>Reviewers need to adapt to new evaluation criteria. |
| **4.5 Implementing more binding data sharing requirements** | Prerequisites for more binding policies: researchers' competencies, availability of infrastructures, funders' ability to monitor compliance. <br>Monitoring compliance could come with burdens and expenses. <br>Funders perceive their own scope of action to be limited by academic freedom ("Wissenschaftsfreiheit"). <br>If data sharing considered as a part of good scientific practice, more binding requirements could be compatible with academic freedom. | Academic freedom is important and a potential limitation for binding data sharing policies. <br>Funders have a right to require data sharing and their funding policies could be more binding. <br>The role of data sharing in good scientific practice must be clarified. |

funders generally described the implementation of policies as a major challenge due to the complexity of this undertaking, the inertia of large funders' organisational structures, and disagreements within funding agencies. International funders reported similar hurdles [35, 39]. While there are no easy solutions, German funders could better overcome these hurdles by coordinating their policy developments and following the policy best practice of international funders more closely.

In our interviews, funder employees perceived the actual and/or potential role of German funders in promoting data sharing to be facilitators and moderators, helping to establish research data infrastructures and trying to raise awareness for data sharing. Some funders emphasised the need for a general cultural change towards more data sharing in science, but also admitted that they prefer to wait for scientific communities to make progress in this area rather than pushing for the cultural change themselves. A similar reluctance was also observable among international funders [39]. In contrast, a number of stakeholder organisation employees felt that funders could and should play a more active role in promoting this cultural change. While it is clear that funders cannot bring about cultural change all by themselves, the

calls for funders to play a more active role are in line with other work that considers funders as (potentially) important promoters of data sharing [6, 20, 21, 27–37]. Our interviews mention several approaches that would allow (especially larger) funders to play a more active role in promoting data sharing, e.g., by providing guidance through their policies, clarifying legal framework conditions and engaging in joint efforts with other stakeholders.

In terms of concrete measures to promote data sharing, our interviews mention the existence and importance of financial incentives or compensation for costs and necessary resources associated with data sharing. However, beyond these financial compensations, there appears to be a lack of other incentives. The wide lack of incentives is both noteworthy and problematic, as numerous studies highlight the importance of incentives for data sharing [5, 45, 72, 75–80]. Similar to the international funder landscape [39], German funders appear to be very aware of the importance of incentives and the current lack thereof. Complementing financial compensations, several German funders strongly consider or already plan to introduce evaluation metrics that recognise data sharing efforts more strongly, including, for example, open science track records.

However, our interviewees also reported various barriers to incentivising data sharing known from other studies [35, 39, 43, 44, 47, 72, 81]. These include internal disagreements within funding agencies with regard to introducing new evaluation metrics that consider data sharing more strongly or convincing grant reviewers of applying those new evaluation metrics. The future implementation of incentives must also take into account that there are also disincentives inherent to the current system of scientific research, for example if researchers perceive data sharing as a competitive disadvantage and additional workload. As potential approaches to address these hurdles concerning the introduction of incentives, interviewees proposed that funders highlight two aspects: First, to increase the intrinsic motivation of researchers, funders should highlight the benefits of data sharing. For example, funders could emphasize that Data Management Plans are not just a bureaucratic burden, but also a helpful and quality-assuring part of the research process. Second, funders could put more emphasis on data sharing as a part of good scientific practice in cooperation with and support from the scientific community. Additionally, German funders could take a closer look at international best practice examples for incentivising data sharing [82–84].

A few interviewees stated that incentives alone may not be sufficient to motivate researchers to share data and that policies with concrete data sharing requirements are also needed. Some funders are considering implementing more binding elements or concrete expectations regarding data sharing, such as obligations for researchers to submit a DMP or to adhere to the FAIR principles. However, and very importantly, two factors were mentioned as challenges to introducing additional requirements into funders' data sharing policies: certain prerequisites and the right to academic freedom. First, some funder employees stated that there are critical prerequisites for a legitimate introduction of more stringent requirements: a) researchers need to know how to share data (competencies); b) researchers need to have functional data sharing infrastructures available; and one interviewee added that c) funders need to be able to monitor grantees' compliance with funders' data sharing requirements or whether researchers actually adhere to the Data Management Plans they submitted. Second, our interviews suggest that German funders are worried about infringing on researchers' academic freedom by imposing more binding and/or additional requirements and therefore perceive academic freedom as a constraint to such efforts. This aspect seems to be particularly noteworthy as interviews with international funding agencies did not reveal academic freedom to be an equally important topic in comparison [39].

This difference in the perception of academic freedom as potential hurdle for the introduction of mandatory data sharing requirements raises questions that, for different reasons, we

can only touch on rather briefly. Does academic freedom have a particularly high status in Germany? Historically, liberal positions in constitutional law have put a strong focus on academic freedom as basic constitutional individual right against state interferences ever since 1848. The German constitution after World War II even strengthened this position further as a reaction to totalitarian negation of academic freedom [85]. In contrast, other constitutional laws such as the one of the US do not explicitly recognize a right to academic freedom [20]. Our interviews indicate that academic freedom seems to be very present in the German academic culture and in science politics. The apparent differences in the perceptions of academic freedom between Germany and the international landscape also seem to related to different perceptions of data sharing and its place in good scientific practice [20]. This is a point we will return to in the following, since it was also expressed by some of our interviewees, stating, for example, that academic freedom should not protect a right to "bad" scientific practice. Yet, the sentiment that data sharing is a part of good scientific practice might not be as present in Germany as it is in other countries.

Judging from our interviews and the underlying exploration of the German research data landscape, progress is being made concerning some of the aforementioned prerequisites for more binding requirements. As to prerequisite a), there appears to be increasing awareness of data sharing reflected in the standards of several disciplines and in the research data policies of research performing organisations [60–62]. This is also reflected in the continuous development of research data infrastructures (b) developed at these organisations. For both of these prerequisites, the development of the NFDI can also be considered as a major contribution. In contrast, a potential monitoring of compliance (c) with funders' data sharing policies or submitted Data Management Plans still seems to be largely absent even with those funders that have such policies in place. International funders also reported monitoring to be a challenge, for example because it requires additional resources [39]. Nevertheless, monitoring data sharing can be an important tool to reduce rule violations and freeriding, identify challenges for funded researchers and provide valuable evidence for funders to improve their policies.

With regard to the perceived conflict between data sharing requirements and respect of academic freedom, our interviewees indicated paths that undermine the straightforwardness of such a conflict or that can substantially mitigate the conflict between more binding data sharing requirements and academic freedom. In the following, we shortly summarize and follow up on these indicated paths. 1) Continuous and progressing investments in and developments of competencies, infrastructures and support for data management and sharing increasingly reduce the costs and burdens of data sharing for individual researchers and thereby reducing one aspect of potentially infringing effects of binding data sharing policies on academic freedom. 2) More binding data sharing policies could (and should) still include legitimate exceptions from data sharing requirements, for example for ethical, legal or commercial reasons and provide embargo periods that protect researchers' interests). 3) Some interviewees from funders and stakeholder organisations pointed out that if the scientific community recognises data sharing as part of good scientific practice, stricter requirements to share data would increasingly be justified. Since the DFG code of conduct presents data sharing for reproducibility and reuse as good scientific practice [64], this condition for stricter requirements might be met soon. However, to what extent and how exactly data sharing is seen as part of good scientific practice in specific scientific communities is a desideratum of future research. 4) As public funding agencies distribute public money and publicly funded researchers receive public money, both have a responsibility to promote the public interest and benefit the scientific communities by promoting and implementing data sharing. In our interviews, some employees from both public and private funders embraced the responsibility to promote the public interest and perceived data sharing to be beneficial in fulfilling this responsibility. In contrast to

public funders, private funders did not refer to the "public-money-reasoning", which is not surprising given that private funders do not spend public money.

We conclude the discussion section with some remarks from the perspective of research ethics. Most basically, from the perspective of research ethics, data sharing should be considered part of good scientific practice and research integrity [19, 21, 28, 34]. The DFG itself points out that funders are responsible for "establishing and maintaining standards of good research practice. Through the design of their funding programs, they create a framework that promotes research integrity" [64]. Public funding agencies have the moral obligation to promote data sharing since they have the obligation to promote good scientific practice and to shape their actions and policies in a way to maximize the potential scientific progress and social benefit from spending public money [21]. The DFG and the BMBF are currently involved in large public funding programmes such as the NFDI to establish research data infrastructures, which raises the question what the DFG and the BMBF can and should do to ensure that these infrastructures for sharing data are actually being used by funded researchers. All these ethical considerations speak in favour of public funding agencies engaging in an active role to promote data sharing through their policies, i.e., by introducing incentives, and speak against funding agencies assuming a "wait-and-see attitude" towards a cultural change in the direction of data sharing. Following the aforementioned remarks from research ethics, one might even come to the conclusion that public funding agencies may or should embark on a path towards more binding data sharing polices.

## 5.1 Limitations

The findings of this study come with some limitations that are common for the methodology of qualitative interview studies. First, the scope of this article is limited both by the size and the composition of our interview sample. We therefore do not claim to offer a comprehensive or representative perspective on the German funder landscape as a whole, but stress that our results are geared towards more important and advanced funders and knowledgeable experts in stakeholder organisations. In addition, our interview results may be influenced by the different response rates among the types of organisations we approached. For these reasons, we cannot rule out a certain bias in the selection. Second, although we used interview guidelines, there were differences in the interview situations. As a result, many of the interview utterances do not allow for exact comparisons and precise quantitative statements (such as "X out of Y interviewees said Z") are very difficult. Third, the expert interview method assumes that "experts" are knowledgeable about their respective fields. However, as it turned out that German funders often do not have data sharing policies in place, several of the interviewees' statements were more hypothetical than based on practical experiences. Fourth, our interviews aimed to collect internal insights and background knowledge, which is why we do not report official statements. Nevertheless, we assume that our interviewees represent the perspective of their organisations to some extent. Finally, our interviews took place in 2021 and 2022. As data sharing policies of funding agencies are constantly evolving, changes may have been made in the meantime.

## 6. Conclusion

Leading international funding agencies have introduced explicit data sharing policies to promote the sharing of research data. However, a closer look at funding agencies in Germany shows that data sharing policies are still an exception there. The goal of this article is to provide an interview-based account of the status quo of German funders' data sharing policies, their efforts to promote data sharing, and prevailing barriers for advancing data sharing. Based on

the data collected in 16 interviews with research data and funding policy experts from these organisations, we arranged our findings in five result sections: Interviewees' perspectives on the status quo of German funders' data sharing policies, their perspectives on the role of funders in promoting data sharing, their perspectives on funders' current and potential measures to promote data sharing, their perspectives on general barriers and limitations and their perspectives on implementing additional data sharing requirements. Our findings suggest that, among other obstacles, especially the high value placed on academic freedom in Germany is a major reason for the slow development of German funders' data sharing policies. While large funders' part in developing a national research data infrastructure is an important contribution to more data sharing, German funders ought to catch up with international funders' best practice, such as designing explicit and clear data sharing policies, providing incentives such as new evaluation metrics, and monitor compliance with their policies to some degree. Lastly, funders should promote data sharing as a part of good scientific practice more strongly.

## Supporting information

**S1 Appendix. Interview guideline for German funders.**
(DOCX)

**S2 Appendix. Interview guideline for stakeholder organisations.**
(DOCX)

**S3 Appendix. Transcription and coding guideline.**
(DOCX)

**S4 Appendix. Category system.**
(DOCX)

## Acknowledgments

The authors would like to thank Maya Doering and Franziska Ziegler for their assistance with the transcription and the coding of the interview material. Furthermore, we want to thank Prof. Dr. Dr. Eva Winkler and the Section *Translational Medical Ethics* at the National Center for Tumour Diseases, Heidelberg, for their valuable support and feedback. We also give thanks to our partners within the joint research project DATABLIC, Prof. Dr. Michael Fehling and Miriam Tormin (Bucerius Law School, Hamburg), Prof. Dr. Christiane Schwieren and Tamás Olah (University of Heidelberg) for fruitful discussions. Last but not least, we want to express our immense gratitude to all interviewees for their participation and contribution to our research.

## Author Contributions

**Conceptualization:** Michael Anger, Christian Wendelborn, Christoph Schickhardt.

**Data curation:** Michael Anger.

**Formal analysis:** Michael Anger, Christian Wendelborn.

**Funding acquisition:** Christoph Schickhardt.

**Investigation:** Michael Anger.

**Methodology:** Michael Anger.

**Project administration:** Christoph Schickhardt.

**Supervision:** Christoph Schickhardt.

**Visualization:** Michael Anger.

**Writing – original draft:** Michael Anger.

**Writing – review & editing:** Christian Wendelborn, Christoph Schickhardt.

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
