## [Decision Letter · Decision Letter 0]

23 Oct 2023

PONE-D-23-25796German funders’ data sharing policies – A qualitative interview studyPLOS ONE

Dear Dr. Anger,

Thank you for submitting your manuscript to PLOS ONE. After careful consideration, we feel that it has merit but does not fully meet PLOS ONE’s publication criteria as it currently stands. Therefore, we invite you to submit a revised version of the manuscript that addresses the points raised during the review process.

We look forward to receiving your revised manuscript.

Kind regards,

Dzintars Gotham

Academic Editor

PLOS ONE

6. Thank you for stating the following in the Competing Interests section: 

[The authors do not have any competing interests. This study is part of a research project which is funded by the German Federal Ministry of Education and Research and reports to the project execution organisation DLR Projektträger. Both organisations were part of our larger field of study (German funders) but had no influence on the design or content of this study.]. 

Reviewers' comments:

Reviewer's Responses to Questions

**Comments to the Author**

1. Is the manuscript technically sound, and do the data support the conclusions?

Reviewer #1: Partly

Reviewer #2: Yes

2. Has the statistical analysis been performed appropriately and rigorously? 

Reviewer #1: N/A

Reviewer #2: N/A

3. Have the authors made all data underlying the findings in their manuscript fully available?

Reviewer #1: No

Reviewer #2: Yes

4. Is the manuscript presented in an intelligible fashion and written in standard English?

Reviewer #1: Yes

Reviewer #2: Yes

5. Review Comments to the Author

Reviewer #1: This article is well structured and addresses an important topic on the role of funders in promoting data sharing in the scientific community. It provides a balanced view both from within the funder community and from external stakeholders and could potentially ignite discussions on formulation and implementation of funder-driven data sharing policies in Germany. I would like to raise a few points on the study design and presentation of the results.

1. The research is premised on the idea that German funding agencies are lagging in the formulation and enforcement of data sharing policies compared to their international counterparts. The article would benefit from a presentation of evidence to support this claim. For instance, the assertion in Line 121, "However, we only identified a few funders that have some kind of data sharing policy in place," can be substantiated by references or brief descriptions of the specific data sharing policies that these "few funders" have implemented. It is essential to also distinguish between comprehensive data sharing policies and general guidelines, such as the BMBF's "Action Plan on Research Data" and the DFG's "Code of Conduct for Safeguarding Good Scientific Practice".

2. The use of guided interviews to collect qualitative data is appropriate for this research. However, the article does not provide sufficient detail regarding sample selection. The first criterion for sample selection is based on importance of the funding agency in terms of annual funding. The second criterion is based on size of the funder. In both instances it is important to clarify what parameters or thresholds were used to determine the ‘importance’ and ‘size’ of the funding agencies.

3. The variability in response rates among different groups (funders, stakeholders and Projektträger) may introduce selection bias which could potentially influence the study results. I encourage the authors to provide a breakdown of interviewees based on their affiliation, as outlined in Fig 1: Sampling Procedure. Additionally, it will be beneficial to acknowledge the potential for selection bias as a study limitation.

4. Public funders are perceived to have a responsibility to promote data sharing for public good (line 870-873). Are these sentiments shared by the funders themselves? What does the data say with regard to differences in expectations for public and private funders?

5. This work highlights a notable tension between academic freedom and data sharing policies. Academic freedom is perceived as an obstacle for German funders in introducing mandatory data sharing requirements. This conflicts findings from other countries (line 839-841). A more thorough exploration of this unique dynamic in the discussion section would significantly enhance the study.

7. Minor suggestions:

Line 23: Define ‘other organisation’ more precisely

Line 31: Clarify ‘Barriers’... to what?

Line 58: Support the claim that “Several leading international funding agencies have introduced explicit policies on data sharing” with a reference

Reviewer #2: The authors conduct a qualitative assessment of funder and funding stakeholder perception of German funder data sharing requirements and find important insights into internal perspectives, obstacles, and potential solutions for promoting data sharing in German funded projects. While I am not a social scientist or very familiar with qualitative methods, the data are well defined and lessons are clearly summarised and attributed to data collected (interview excerpts). I just have a few clarifying questions and language to be address

Line 58: briefly define FAIR data

Lines 62 – 65: Comparisons of German practices to other international funders is not an objective of the study so I suggest downplaying this gap and promoting the perceptions piece. The way it is worded the perceptions are secondary.

Line 118: “increased political awareness” compared to what? Previous years? Please clarify

Line 120: Please provide some examples of open science policies developed

Line 121: “identified a few funders” out of how many reviewed/investigated?

Line 145: change important to “top” or “influential”

Line 231: “consideration of research data” please explain in more detail. How is research data considered?

Line 232: “implulses” awkward wording: Motivate, accelerate, inspire?

Line 467: change “is” to “are” this is a quote so not sure if the edit is appropriate.

Line 682: Define or explain “academic freedom” as it pertains to German law for readers that are unfamiliar with the specifics

6. PLOS authors have the option to publish the peer review history of their article (what does this mean?). If published, this will include your full peer review and any attached files.

Reviewer #1: No

Reviewer #2: No

---

## [Author Response · Author response to Decision Letter 0]

7 Dec 2023

Responses to the comments by the academic editor and the reviewers 

First of all, we want to thank the editor and the reviewers for helping us to improve our research article.

In the following, we respond to each point raised by the editor and the reviewers. For detailed insights into the changes to our manuscript, please also take note of the marked-up copy of the manuscript with changes in “track changes”-mode. In this document, all of our references to lines where we made changes refer to the revised manuscript with track changes.

Sincerely, the authors

Responses to the Academic Editor

Comment A: Thank you for submitting your manuscript to PLOS ONE. After careful consideration, we feel that it has merit but does not fully meet PLOS ONE’s publication criteria as it currently stands. Therefore, we invite you to submit a revised version of the manuscript that addresses the points raised during the review process.

Response: Thank you very much for this comment and for considering our manuscript for publication in PLOS ONE. We used your valuable remarks to improve our manuscript. We hope that our revised submission now meets PLOS ONE’s publication criteria. With this letter, we also submit a marked-up copy of our manuscript and an unmarked version of our revised manuscript.

Comment B: 1. Please ensure that your manuscript meets PLOS ONE's style requirements, including those for file naming.

Response: Thank you for pointing out the need to improve compliance with PLOS ONE’s style requirements. We reviewed our manuscript and the supporting files and implemented the following style changes:

 We moved all captions for our supporting information files to the very end of our manuscript after the reference list. Our in-text citations of the captions have also been updated to match the captions at the end of the document (all of them are now named S1 Appendix – S3 Appendix).

 We made some changes to the content provided by Fig1, following the suggestions of reviewer 1.

Comment C: 2. In your Data Availability statement, you have not specified where the minimal data set underlying the results described in your manuscript can be found. PLOS defines a study's minimal data set as the underlying data used to reach the conclusions drawn in the manuscript and any additional data required to replicate the reported study findings in their entirety. All PLOS journals require that the minimal data set be made fully available. Upon re-submitting your revised manuscript, please upload your study’s minimal underlying data set as either Supporting Information files or to a stable, public repository and include the relevant URLs, DOIs, or accession numbers within your revised cover letter. 

Response: Thank you for raising this point. Yet, after double-checking the Data Availability statement we submitted, we are still under the impression that we are fully compliant with this requirement. The Data Availability statement we submitted provided the following information: 

“This article reports findings from expert interviews with employees of funding agencies and stakeholder organisations. The full transcripts of the interviews contain highly sensitive qualitative data – including person-related data – and are therefore subject to data protection law, particularly the EU-GDPR. Consent for participation in the interview study was obtained on the condition of complete anonymity and confidentiality. The interviewees’ trust in strict anonymity and confidentiality allowed us to gain specific insights that constitute the value of our data (such as information and opinions about their work and the internal situation of organisations which are also their employers), but at the same time make an unlimited publication of the interview transcripts impossible for legal and ethical reasons. To enable replication of the findings reported in this article without violating our legal and ethical obligations, we created excerpts of the interviews. Since we cannot fully exclude the possibility that even the excerpts might still contain some information that could technically allow third parties to re-identify the interviewees, we grant access to the data solely upon request. This minimal data set is deposited in the heiDATA repository, DOI: https://doi.org/10.11588/data/FJAG0X. Ethical clearance was obtained – under the above-mentioned data protection conditions – from the Data Protection Office of the German Cancer Research Center, Heidelberg (registration no. D120-P989). Please address data access and usage requests to use-and-access@med.uni-heidelberg.de. Requests for data use and access will be reviewed and decided upon by the independent Data Use and Access Committee of the Heidelberg University Hospital and the authors.” 

With this statement, we do specify where our minimal data set can be found by naming both the repository where it is uploaded and, more importantly, the digital object identifier that directly links to our data set (https://doi.org/10.11588/data/FJAG0X). heiDATA is a public open research data repository run by Heidelberg University (https://heidata.uni-heidelberg.de/). We included this information in our revised cover letter and also made minor changes to our data availability statement to better reflect this information.

Unfortunately, we cannot share all data generated by the expert interviews, since sharing the full transcripts of the interviews would violate data protection and endanger the confidentiality and anonymity that we granted to the interviewees as part of our consent form. Therefore, we only share a minimal underlying data set. A more detailed explanation of our approach and data protection measures, which are grounded on ethical and legal restrictions to sharing our data publicly, follows in the next segment (Commend D). 

Comment D: Important: If there are ethical or legal restrictions to sharing your data publicly, please explain these restrictions in detail. Please see our guidelines for more information on what we consider unacceptable restrictions to publicly sharing data. Note that it is not acceptable for the authors to be the sole named individuals responsible for ensuring data access.

Response: Thank you for pointing to your guidelines and giving us the space to explain the restrictions to our data sharing efforts in detail.

There are indeed strong legal and ethical restrictions to sharing our data publicly. The interview transcripts are sensitive data and to be considered personal-related data according to the European Union’s GDPR. Data protection statements are particularly high in Germany and we also have to adhere to additional regulations, such as the data protection law by the state of Baden-Wuerttemberg (“Landesdatenschutzgesetz”).

The transcripts of each interview contain lots of personal information, data and statements about the interviewees and their institutions. Some of these data and information and personal statements are highly sensitive, for example when interviewees talk about their own superiors and the working environment of organisations that are also their employers. In our consent form and before starting each interview, we guaranteed each participant confidentiality and anonymity, allowing them to make statements that were sometimes critical of their respective organisations (and therefore, their employers) and that might displease superiors, colleagues, or other stakeholders. Some of our interviewees were particularly careful about the protection of their privacy and were only willing to be interviewed on the condition of full anonymity.

We collected and processed the interview data on the legal basis of informed consent. This consent does not contain any permission to share our transcripts with third parties or to make the data publicly available. The interviewees were assured that all statements would be treated as confidential. Full anonymity was guaranteed to the interviews, which can only be warranted as long as their statements would never be traceable back to their organisations, which therefore also cannot be disclosed in detail. These conditions are at the core of our registration of the processing of personal-related data that we submitted to the data protection office of our research institution (German Cancer Research Center, registration no. D120-P989). As a consequence, sharing the transcripts publicly would violate good research practice, our promises made to the interviewees, the interviewees’ trust in us and, last but not least, our obligations as employees towards the data protection office of our employer. 

However, we also believe in Open Science, transparency, and reproducibility. Therefore, we have been looking for a compromise that could respect our legal, ethical and scientific duties on the one hand and, on the other, address the demand for sharing data to enable replicability of our research results. Following the PLOS ONE recommendations, we created extensive excerpts of the interviews to be made available in a secure and controlled manner upon request. The interview excerpts contain all statements that are essential for our research findings. We, the authors, carefully checked all interview excerpts in order to prevent them from containing information that could allow third parties to (directly) identify the interviewee or her organization. We would like to draw your attention to the necessity that the interviewees, whom were granted anonymity, must not even be identifiable from the excerpts to their own colleagues, who may have additional information on the interviewees (and could use their additional information to attempt to identify the interviewees). As mentioned in our article, we conducted interviews with open science/research data contacts from leading German funding agencies and so-called stakeholder organisations, which already leads to a limited sample size. As consequence, a number of our interviewees are at risk of re-identification if the organisation in question would be revealed. 

Furthermore, and despite our considerable efforts to anonymise our interview excerpts, qualitative data material is very hard to fully anonymise while at the same time preserving its scientific value. Given these necessities and conditions, we felt that the only suitable and responsible way to share the excerpts is to upload them in heiDATA, a secure repository with restricted access and share them upon request (for more details about heiDATA, please see our response to Comment C). Considering the sensitivity of our data, the applying data protection laws, the conditions to which the interviewees consented, and professional obligations of good scientific practice, we can only share a minimal data set, consisting of the interview excerpts, via restricted access.

For data access requests we entrust the Use & Access Committee of the Heidelberg University Hospital. Since it combines experience in handling sensitive data, commitment to data sharing and works with established criteria and processes, we consider the Use & Access Committee to be an appropriate institutional contact for other researcher’s data access requests.

Data access requests can be sent to the Use & Access Committee of the Heidelberg University Hospital under the following email address: use-and-access@med.uni-heidelberg.de

Comment E: 4. We note that you have stated that you will provide repository information for your data at acceptance. Should your manuscript be accepted for publication, we will hold it until you provide the relevant accession numbers or DOIs necessary to access your data. If you wish to make changes to your Data Availability statement, please describe these changes in your cover letter and we will update your Data Availability statement to reflect the information you provide.

Response: As stated in our Data Availability statement, our minimal data set underlying our results can be found at heiDATA, the Open Research Data repository of Heidelberg University (https://heidata.uni-heidelberg.de/). heiDATA is a public repository and also registered with re3data (https://www.re3data.org/), therefore ensuring pertinent metadata standards. Our dataset has the following DOI: https://doi.org/10.11588/data/FJAG0X, which is mentioned in the Data Availability Statement.

With regard to the access to our data, we explained the restrictions on data access in the response to the previous comment (Comment D).

As far as we are aware, our Data Availability Statement already reflects all necessary information required by PLOS ONE’s data sharing guidelines. We decided to make some minor changes by improving the structure of the statement and by deleting three words, clarifying that the Use & Access Committee is responsible for granting access. 

Our revised data availability statement now reads as follows:

“This article reports findings from expert interviews with employees of funding agencies and stakeholder organisations. The full transcripts of the interviews contain highly sensitive qualitative data – including person-related data – and are therefore subject to data protection law, particularly the EU-GDPR. Consent for participation in the interview study was obtained on the condition of complete anonymity and confidentiality. The interviewees’ trust in strict anonymity and confidentiality allowed us to gain specific insights that constitute the value of our data (such as information and opinions about their work and the internal situation of organisations which are also their employers), but at the same time make an unlimited publication of the interview transcripts impossible for legal and ethical reasons. To enable replication of the findings reported in this article without violating our legal and ethical obligations, we created excerpts of the interviews. Since we cannot fully exclude the possibility that even the excerpts might still contain some information that could technically allow third parties to re-identify the interviewees, we grant access to the data solely upon request. This minimal data set is deposited in the heiDATA repository, DOI: https://doi.org/10.11588/data/FJAG0X. Ethical clearance was obtained – under the above-mentioned data protection conditions – from the Data Protection Office of the German Cancer Research Center, Heidelberg (registration no. D120-P989). Requests for data use and access will be reviewed and decided upon by the independent Data Use and Access Committee of the Heidelberg University Hospital. Please address data access and usage requests to use-and-access@med.uni-heidelberg.de. 

If we did miss something essential by honest mistake, we are more than happy to provide any missing information.

Comment F: 5. Please include captions for your Supporting Information files at the end of your manuscript, and update any in-text citations to match accordingly. Please see our Supporting Information guidelines for more information: http://journals.plos.org/plosone/s/supporting-information. 

Response: Thank you for pointing this out. We moved all captions for our supporting information files to the very end of our manuscript after the reference list. Our in-text citations of the captions have been updated accordingly (now named S1 Appendix – S3 Appendix).

Comment G: 6. Thank you for stating the following in the Competing Interests section: 

[The authors do not have any competing interests. This study is part of a research project which is funded by the German Federal Ministry of Education and Research and reports to the project execution organisation DLR Projektträger. Both organisations were part of our larger field of study (German funders) but had no influence on the design or content of this study.]. 

Please confirm that this does not alter your adherence to all PLOS ONE policies on sharing data and materials, by including the following statement: “This does not alter our adherence to PLOS ONE policies on sharing data and materials.” […] Please include your updated Competing Interests statement in your cover letter; we will change the online submission form on your behalf.

Response: Thank you for providing us detailed information on how to improve the accuracy of our Competing Interests statement. We hereby confirm that the circumstances described in our statement do not alter our adherence to PLOS ONE policies on sharing data and materials.

We therefore added the phrase “This does not alter our adherence to PLOS ONE policies on sharing data and materials” to our Competing Interests statement as suggested by the Academic Editor. 

Furthermore, we included our updated Competing Interests statement in our revised cover letter. Thank you for changing the online submission form accordingly.

Responses to the reviewers

Reviewer #1

Reviewer#1 Comment A: This article is well structured and addresses an important topic on the role of funders in promoting data sharing in the scientific community. It provides a balanced view both from within the funder community and from external stakeholders and could potentially ignite discussions on formulation and implementation of funder-driven data sharing policies in Germany. I would like to raise a few points on the study design and presentation of the results.

Response: Thank you for your positive feedback and highlighting the importance of our findings. We are very grateful for your helpful comments and want to address your remarks in the following.

Reviewer#1 Comment B: 1. The research is premised on the idea that German funding agencies are lagging in the formulation and enforcement of data sharing policies compared to their international counterparts. The article would benefit from a presentation of evidence to support this claim. For instance, the assertion in Line 121, "However, we only identified a few funders that have some kind of data sharing policy in place," can be substantiated by references or brief descriptions of the specific data sharing policies that these "few funders" have implemented. 

Response: Thank you for pointing out this pathway to improve of our article. 

From our point of view, the idea that German funders are lagging behind rest on different observations and evidence. First, there is the actual lack of evidence for German funders’ data sharing policies – only three funders in our sample had data sharing policies on their website. We now explicitly name those funders and refer to their documents. Second, and in addition to this quantitative aspect mentioned before, there is a qualitative aspect to our observation: even for those German funders who have (as we explicitly say) “some kind of” data sharing policies in place (like the DFG and the BMBF), those policies are not always comprehensive or explicit (we will address this topic further for a later comment of yours). This assessment is, of course, a relative one and can, third, only be fully understood by comparison with international funders. As we did a study on international funding agencies’ data sharing policies (see reference no. 41), it became obvious to us that German funders’ policies are not as advanced. Finally, a few German funders (among them one of the leading ones!), admitted that they perceive themselves as lagging behind in international comparison (line 320-323), while most funders admitted that they are either still at the stage of considering or planning the development of data sharing policies. 

To improve our presentation of evidence, we took the following steps: A) We clarified in the introduction that German funders’ policies seem neither very advanced nor widespread (see reference 44). B) In line 130-132, we specified that we found only three funders with some kinds of policies on their websites (BMBF and DFG and a third funder we refrain from naming and describing explicitly for reasons of data protection). C) In line 153-158, we added a summarizing formulation according to which we only identified three funders with documents that qualify as policies in a broad sense, but which do not appear advanced in international comparison (see reference 41 and 44), which tend to have explicit and focussed data sharing policy documents on their websites.

Finally, we would like to highlight that we report further evidence to support this claim in our results section and draw comparisons to leading international funders in our discussion.

We hope these changes do sufficiently meet your concern.

Reviewer#1 Comment C: It is essential to also distinguish between comprehensive data sharing policies and general guidelines, such as the BMBF's "Action Plan on Research Data" and the DFG's "Code of Conduct for Safeguarding Good Scientific Practice".

Response: You are addressing a very important point here which we, the authors, were also discussing in the process of conducting our study and writing this article. From our point of view, it leads us to the question which documents and guidelines qualify as (comprehensive) data sharing policies? But it also raises follow-up questions such as: What are the functions and criteria that are constitutive for (comprehensive) data sharing policy? Are mere guidelines and codes of conduct also policies? Although our article is mostly empirical, we fully acknowledge the importance of these conceptual questions.

Allow us to talk about our definition of data sharing policies first. In the introduction to our research article, we provide a definition of data sharing policies (“In this paper, we use the term “data sharing policies” to describe a framework of grant conditions, instructions, incentives, evaluation mechanisms and monitoring criteria regarding data management and sharing”). This definition is not the be-all and end-all of defining data sharing policies, but it at least gives the readers some idea of what we understand by data sharing policy. We admit that this definition is very broad in its scope, but we felt like this would be the most suitable for our empirical study.

We find it important to note, and this leads us back to your previous comment (Comment B), that there are different levels of policy development and some funders are more advanced than others. We concede that it is debatable to describe the documents you refer to as policies (i.e., the BMBF’s Action Plan and the DFG Code of Conduct), which is why describe them in the text as “some kind of policies” and add that there are additional documents to supplement them. We now summarize in the manuscript (line 154-156) that we consider the combination of the DFG’s and the BMBF’s general guidelines and their more specific instructions to be data sharing policies in a broad sense of the term. We also think that this is an important point in itself, as it stresses our observation that German funders are lagging behind in international comparison – a point we also clarify in the manuscript (line 156-158).

Overall, we do agree with your position that a distinction like this important and necessary. Judging, from both an ethical point of view and our socio-empirical research, a distinction like this can actually be very tricky for the following reasons:

First, at least to us, the term “comprehensive data sharing policies” could be understood as an evaluative, or even normative description, because it suggests a certain functional quality of a data sharing policy. From our point of view, it seems more difficult to define what a comprehensive data sharing policy is than what a data sharing policy is, which is why we prefer a broader and more neutral term.

Second, it is important to note (as we did in line 135-140), that the BMBF complements its Action Plan on Research Data with specific topic-related funding programs, to which research projects can apply in order to receive funding. These programs implement the Action Plan on Research Data to some extent, e.g., by requiring future grantees to make their data FAIR. In this sense, and although the concrete requirements verify from funding program to funding program, these calls also fulfil some aspects of a policy framework which, in addition to the Action Plan, may qualify as a policy in a broader sense, depending on the definition. 

Third, the DFG supplements its Code of Conduct with additional guidelines on research data (line 148-153), which are more concrete and, together with the Code of Conduct, may qualify as a policy in a broader sense. 

Fourth, in our interviews, our interviewees widely regarded the aforementioned documents as actual data sharing policies. On top of that, those documents were considered to be essential points of references for the policy development of smaller agencies. 

From our point of view, this topic is, at the centre, a question of defining data sharing policies. A definition of a (comprehensive) data sharing policy that follows advanced, clear, and explicit data sharing policies like the ones of leading international funding agencies, might not consider the documents provided by leading German funders to be data sharing policies. But, at least to us, it does seem that the documents by the BMBF and DFG provide a (very general) framework that at least fulfils some functions of a policy, such as providing general instructions and explanations, raising awareness to incentivites for sharing data, and referring to changing evaluation metrics.

Following your important comment, we made three changes to our manuscript. First, we added another sentence to our definition of data sharing policies, stating: “In practice, there are different stages of policy development, but we would expect an advanced data sharing policy to encompass all these aspects to some degree.” [line 58-62]. This way, we want to integrate your point and highlight potential deficits at the same time. Second, we added a short summary to chapter two, stressing that we only found three founders that have a data sharing policy in a broad sense. Yet, these documents are not as advanced as the ones of leading international funders. We are not under the impression that all leading funders have comprehensive data sharing policies either (see reference 41), but we hope this adds some important clarification. Third, we made some minor changes to our discussion section (line 804-805), stating that some larger funders in German have Data Sharing Policies in a broad sense of the term, but that the policies of leading international funders tend to be more comprehensive.

Overall, we hope that you find our approach to be plausible. Thank you for improving both our definition and the precision of our article as a whole with this comment. 

Reviewer#1 Comment D: 2. The use of guided interviews to collect qualitative data is appropriate for this research. However, the article does not provide sufficient detail regarding sample selection. The first criterion for sample selection is based on importance of the funding agency in terms of annual funding. The second criterion is based on size of the funder. In both instances it is important to clarify what parameters or thresholds were used to determine the ‘importance’ and ‘size’ of the funding agencies.

Response: Thank you for pointing this out. We do agree that the information we provide regarding our sample selection could be more detailed and provide a clearer picture of our sampling process, which is why we want to add information to our manuscript. 

For public funding agencies, we now state more explicitly that we turned to the DFG, the BMBF, and other federal ministries spending money on research funding and stress that they are the most important funders in Germany, which holds true both for the amount of funding (where DFG and BMBF stand heads and shoulders above other federal ministries), but also with regard to their political significance [line 163-165]. By adding to the text that we approached the most important funders, we hope to provide more clarity that we started our contacting process “at the top” of Germany’s funding organisations. The threshold, in this sense, is therefore belong to the list of the DFG or the federal ministries that publicly fund research. For this reason, we removed the information about state ministries in the text, as we just started contacting ministries on this level to find more interviewees after some negative responses or non-responses from federal ministries. We concede that the political significance is difficult to quantify and relies mostly on our estimation, but to our knowledge, there are no other public funding agencies of comparable significance in Germany. We hope that with this additional explanation and our changes to the manuscript we now better explain the sample criterion “importance” as an overlapping combination of annual funding and political significance, which mostly points to federal ministries.

For private funding agencies (usually charities), providing a concrete threshold is more complicated. At least to our knowledge, there is no exhaustive overview of the concrete amounts of funding of all German private funders, making it difficult to determine which are the largest private funders in Germany. In our sampling process, we turned to the more prominent and pertinent private funders of health research for two reasons: first, and based on our affiliation and field, we have a better overview of what the more important private funders in this area. Second, health research is also considered to be more advanced in the area of data sharing (see reference no. 41), which led us to believe that our interviews could gather more data here. We admit that our initial description (“larger”) is not ideal here, as it indicates a selection based on the size of the funders, suggesting a selection based on annual funding or organizational size of the funders. We changed the article to better reflect this information [line 165-169], stating that we approached some of the important charities that fund health research. 

We agree with your assessment that our article needs more details about sampling selection. Following your remarks, we now provide more concrete information about the sampling process in general, starting with stating the sample size of interviews we were aiming for, the composition of our preliminary sample, and some details about how we adjusted the sample after some negative responses [line 177-182]. We also used this information to update Figure 1. Additionally, we made the aforementioned adjustments to our wordings to better explain our sampling process and move away from expressions that indicate mostly quantitative metrics for our selection, as we also had qualitative criteria in place (e.g., funders of health research). 

Reviewer#1 Comment E: 3. The variability in response rates among different groups (funders, stakeholders and Projektträger) may introduce selection bias which could potentially influence the study results. I encourage the authors to provide a breakdown of interviewees based on their affiliation, as outlined in Fig 1: Sampling Procedure. Additionally, it will be beneficial to acknowledge the potential for selection bias as a study limitation.

Response: You are right to raise this important methodological aspect, and we are grateful for your helpful suggestion. We are aware of the risk of selection bias in studies such as ours. Since complete representativeness is not a goal of qualitative research and cannot be achieved through non-random or incomplete selection, we cannot rule out selection bias for our article. 

In the following, we would like to justify our methodological approach in light of this problem and explain how we address it in our revision. The general goal of our study is to gain insights into the state of the art of German funders’ efforts to promote data sharing via their policies. In doing so, we attempted to gather qualitative data by talking to employees of leading funders as well as stakeholder organisations invested in the topics of research data sharing. This way we are able to provide some inside knowledge, but do not claim to offer a representative account in this area.

By conducting expert interviews, we aimed at approaching knowledgeable experts for the topic of research data, data sharing, and data sharing policies, in order to obtain valuable inside information about their perceptions and insights into their organisations’ efforts. In line with the goal of our study and our method, our sample selection is focused on funders and stakeholder organisations that are large, influential, or can be considered advanced, but this was an intended and reasoned approach of our sampling strategy. The reason for this approach is twofold: First, we expected and perceived those funders to have more knowledge about data sharing policies, resources dedicated in this area, if they do not already have policies like this themselves. Second, we assumed bigger funders to have an impact on the German funding landscape at large and to have a role model function for smaller funders. 

With this goal and method in mind, our study and its results do not claim to be representative of the German funding landscape at large. It primarily focusses on leading German fund agencies and knowledgeable experts from stakeholder organisations. From our perspective, a more random and unbiased sample may have been more representative of the German funder and research data landscape at large, but would have left us with significantly less insights to work with. It may have confirmed our basic assumption that data sharing policies are not very advanced in Germany, but it most likely would not have provided us with insights into the reasons for this development.

We fully agree with you that the different response rates among different groups might potentially influence our study results. While the response rates of public funders (40%) and private funders (50%) are comparable to some degree, the response rates of project execution organisations (12,5%) and stakeholder organisations (100%) are evident outliers. This led us to interview slightly less funders than we hoped for, significantly less project execution organisations than we wanted to talk to, and led us to include more interviews with stakeholder organisations. Because of this, stakeholders’ perceptions have more room than initially anticipated. This is also documented in our results, which we break down into the perspectives of two groups: funders (public, private, and Projektträger) and Stakeholders. 

Following your suggestion, we now provide a more detailed breakdown of the kinds of organisations we approached, including the number of interviews with public and private funders (line 177-182 and line 189-190). As you correctly point out, this procedure is also more consistent with our depiction in Fig1: Sampling Procedure, which we updated accordingly. Furthermore, we now offer more details about our sampling process overall. We hope that by updating the description of our sampling processes within the article (see also our response to your previous Comment D), which now offers more details about the composition of the initial sample and the final sample, we are able to satisfy your request for more transparency. Since we cannot exclude the possibility of selection bias, we added a formulation to the limitation section to acknowledge that our results are potentially influenced by the different response rates among the types of organisations we approached and clarify that our study results do not claim to be representative of the German funding and research data landscape at large. 

We hope that we have adequately considered your comments and we have implemented them appropriately.

Reviewer#1 Comment F: 4. Public funders are perceived to have a responsibility to promote data sharing for public good (line 870-873). Are these sentiments shared by the funders themselves? What does the data say with regard to differences in expectations for public and private funders?

Response: Thank you for raising this question and for your interest in this ethical aspect of our study. 

The sentiment that funders have the responsibility to promote the public interest is widely shared by funders we interviewed, as almost all public funding agencies agreed to this. Their line of reasoning when stating this responsibility usually refers to the use of public money: they feel that since they use public money to fund research and researchers, they, as funders, have a responsibility towards the public, i.e., to press that there is a return of good and benefit to the public. The promotion of data sharing for the public interest per se is addressed less clearly. The sentiment that funders have a responsibility to promote data sharing for the public interest is shared by two German funder interviews explicitly (who are among the more important public agencies). Our interviews with international funders (see reference 41 in the article) yield similar results. 

When it comes to differences in expectations for public and private funders, our data is less insightful here. Judging from our interviews, public funders and private funders formulated the goal of promoting the public interest and perceived data sharing to be beneficial in achieving this goal. In contrast to public funders, private funders did not refer to the “public-money-reasoning” that we explained in the previous paragraph. This is not surprising, however, since private funders do not spend public money. 

Hence, what we can confidently state is that this sentiment was more prevalent among public funders, and added some changes to the manuscript in order to address the point you raised and to provide the readers with more information on the topic.

Reviewer#1 Comment G: 5. This work highlights a notable tension between academic freedom and data sharing policies. Academic freedom is perceived as an obstacle for German funders in introducing mandatory data sharing requirements. This conflicts findings from other countries (line 839-841). A more thorough exploration of this unique dynamic in the discussion section would significantly enhance the study.

Response: Thank you for this remark and your interest in this topic, which is of great relevance for our project at large. We suspect particular perceptions of academic freedom in Germany to be one of the primary causes for German funders’ slower development in the area of data sharing policies. However, since monocausality is very unlikely and our empirical data on this matter is rather limited, we are very careful to draw strong causal conclusions here.

Exploring the relationship and potential tensions between academic freedom and data sharing policies is one of the primary research questions and guided the investigations of our interdisciplinary joint project from the beginning. For the project partners of the law subproject it is of particular importance, as they are concerned that funders putting too strong data sharing obligations on funded researchers might go against constitutional rights, i.e., academic freedom. For this reason, we included the question whether the interviewees perceive a conflict between data sharing policies and academic freedom in all of our interviews with national and international funders.

When asking the question whether interviewees perceived data sharing policies to be in conflict with grantees’ academic freedom, it became apparent that German interviewees (particularly those working for funding agencies) perceived more of a conflict here than their international counterparts, who usually did not perceive any conflict at all (one of them even calling it “typically German”!). These are remarkable differences indeed. From a socio-empirical point of view, this dynamic and discrepancy would deserve an empirical investigation of its own, which could address the interplay between legal frameworks and scientific cultures in more detail. We also refer to an analysis from a German legal point of view that covers this exact topic (reference no. 20) and compares the relationship of data sharing policies and academic freedom in two different countries (Germany, United States of America).

In addition to referring to the findings of said law article more explicitly in our discussion, your comment encouraged us to give this topic more space in our discussion and providing more background information on this matter. Finally, we want to highlight that the topic of academic also received a lot of attention at the end of our discussion section, where we discuss how to mitigate the perceived conflict between stricter funder requirements and academic freedom.

We hope that you find our addition to the discussion appropriate. Thank you for your advice on giving more space to this important topic.

Reviewer#1 Comment H: 

7. Minor suggestions:

Line 23: Define ‘other organisation’ more precisely

Line 31: Clarify ‘Barriers’... to what?

Line 58: Support the claim that “Several leading international funding agencies have introduced explicit policies on data sharing” with a reference

Response: Thank you for these suggestions to improve the precision of our article. Following your suggestions, we implemented the following changes to our manuscript:

a) Line 23: Thank you for your suggestion. We completely understand and agree with your concern with the lack of precision of the wording “other organisations”. However, with regard to the limited word count for the abstract due to the journal’s policies we had to omit some details in the abstract due to space constraints (currently, we are using 300/300 words allowed in the abstract, and we are struggling to make any cuts to the abstract in its current version). We now changed the wordings of “other organisations” to “stakeholder organisations” in the abstract. We are aware that this term is not self-explanatory, but we hope that you find it sufficient that we provide a more detailed description and explanation of this term in the introduction, as well as the sample selection section. We hope that you find our approach to be reasonable given this additional explanation. 

b) Line 31: You correctly point out that it is not entirely clear what these barriers exactly are. We adjusted the wording and now say that we mean barriers to the measures by German funders to promote data sharing, which is the previous category mentioned in this brief enumeration. We hope that this provides some clarification, although the aforementioned limitation regarding the word count in the abstract also applies here. Thank you for your suggestion.

c) Line 58: Following your suggestion, we added four references to support the empirical claim in this passage. 

Reviewer#1 Comment I:

3. Have the authors made all data underlying the findings in their manuscript fully available?

Reviewer #1: No

Reviewer #2: Yes

Response: Finally, we want to address the third standardized question by the journal about the availability of data underlying the findings of the article that you answered with “No”. We want to explain our decision to not make the underlying data publicly available, but only with restricted access.

Unfortunately, we cannot share the original transcripts of our expert interviews. We recognize that openly sharing all the transcripts of our expert interviews would be the best in terms of data sharing, transparency and open science.

However, we cannot share the transcripts since they include sensitive data about the interviewees. In our consent form and before starting each interview, we guaranteed each interviewee confidentiality and anonymity, allowing them to make statements that were sometimes critical of their respective organisations (and therefore, their employers) and that might displease superiors, colleagues, or other stakeholders. Only under these conditions, we were able to recruit a sufficient number of interviewees for our study. 

As mentioned in our article, we conducted interviews with open science/research data contacts from leading funding agencies and so-called stakeholder organisations, which leads to a limited sample size. The interviewees are at risk of re-identification if more detailed data are revealed, for instance data about the organisation in question. Therefore, we have no legal or ethical basis to forward the data to third parties or share it publicly. At the contrary, we have the obligation to protect the anonymity and confidentiality of our interview partners. 

Nevertheless, in order to allow other researchers to examine our research results, we provide a minimal data set consisting of excerpts of the interviews. These excerpts contain all essential elements of each interview and thus comprise all relevant data underlying our findings. Since the excerpts do contain a lot of information and a full anonymization is very difficult in qualitative socio-empirical studies such as ours, there still remains a risk for the anonymity and confidently of the interviewees. To counterbalance this risk, we provide the excerpts only via restricted access upon request. That being said, we’re happy to fulfil requests for data access for the purpose of the review of this study. 

We already stated in our response to the Academic Editor that we deposited our minimal data set in heiDATA, the institutional research data repository of the University of Heidelberg. The minimal data set can be found using the following digital object identifier: https://doi.org/10.11588/data/FJAG0X

Formal data access requests can be sent to the Use & Access Committee of the Heidelberg University Hospital: use-and-access@med.uni-heidelberg.de

We hope that we have implemented all your suggestions and requests to your satisfaction. Thank you again for your valuable feedback!

Reviewer #2

Reviewer#2 Comment A: The authors conduct a qualitative assessment of funder and funding stakeholder perception of German funder data sharing requirements and find important insights into internal perspectives, obstacles, and potential solutions for promoting data sharing in German funded projects. While I am not a social scientist or very familiar with qualitative methods, the data are well defined and lessons are clearly summarised and attributed to data collected (interview excerpts). I just have a few clarifying questions and language to be address.

Response: We are very pleased that you appreciate the importance of the insights of our research article. Before we respond in detail to your specific comments, we would like to thank you very much for your helpful comments. Since we, the authors, are not English native speakers, we also appreciate your remarks to improve the linguistic accuracy of our article.

Reviewer #2 Comment B: Line 58: briefly define FAIR data

Response: We assumed that most readers interested in this topic are familiar with the term “FAIR data” and agree with you that the concept is important enough to be defined briefly. We updated our manuscript accordingly.

Reviewer #2 Comment C: Lines 62 – 65: Comparisons of German practices to other international funders is not an objective of the study so I suggest downplaying this gap and promoting the perceptions piece. The way it is worded the perceptions are secondary.

Response: Thank you for pointing this out. The primary goal of our study is indeed to investigate perceptions of German funders’ efforts to promote data sharing. Therefore, we now add more emphasis on the aspect of perceptions of data sharing policies in Germany in our manuscript. 

In doing so, we also removed the passage about the gap between German and international funders, as it was indeed too popular and also addressed earlier in the text to some degree. That being said, we do believe that the aim and research question of our study unfolds its fullest potential against the background of this apparent gap between German and international funders and could be characterised as a subordinate objective. The observation that German funders’ data sharing policies are lagging behind in international comparison was also one of the starting points of the research project of which this study is a part of. Nevertheless, we agree with your observation that it is too prominent here and might cause confusion about the primary purpose of our article.

We appreciate you raising this point and helping us to specify the key messages of our article. We hope that we are introducing the primary purpose of our study more clearly now.

Reviewer #2 Comment D: Line 118: “increased political awareness” compared to what? Previous years? Please clarify

Response: By using the wording “increased political awareness” we indeed refer to the growing awareness of political actors in recent years, and compared to previous governments. We hope that the adjustments made in the revised manuscript now contribute to a better comprehensibility of this phrase.

Reviewer #2 Comment E: Line 120: Please provide some examples of open science policies developed

Response: Following your suggestion, we added some examples of open science policies as references. Thank you for this idea, as it provides a stronger basis for this empirical statement.

Reviewer #2 Comment F: Line 121: “identified a few funders” out of how many reviewed/investigated?

Response: We clarified that we investigated sixteen funding agencies overall and added this information to our manuscript. Thank you for raising this question.

Reviewer #2 Comment G: Line 145: change important to “top” or “influential”

 Response: Thank you for pointing this out, we updated our manuscript accordingly. 

Reviewer #2 Comment H: Line 231: “consideration of research data” please explain in more detail. How is research data considered?

 Response: We realised that this formulation does not adequately reflect what we are aiming to say, so we changed the phrase to “greater appreciation of research data”.

Reviewer #2 Comment I: Line 232: “impulses” awkward wording: Motivate, accelerate, inspire?

Response: Thank you for hinting at this wording problem. We replaced the word with one of your suggestions.

Reviewer #2 Comment J: Line 467: change “is” to “are” this is a quote so not sure if the edit is appropriate.

Response: Since all the interviews were translated from German to English by the authors, corrections of our translation mistakes are both appropriate and welcome. Thank you for pointing this out, we applied the change as you suggested.

Reviewer #2 Comment K: Line 682: Define or explain “academic freedom” as it pertains to German law for readers that are unfamiliar with the specifics

Response: We added some clarification to this legal concept in the manuscript. To make it more comprehensible for readers who want to know more about the specifics, we also added a reference to the official English translation of the German constitution. 

We hope that we have implemented all your suggestions to your satisfaction. Thank you again for your valuable feedback.

---

## [Decision Letter · Decision Letter 1]

21 Dec 2023

German funders’ data sharing policies – A qualitative interview study

PONE-D-23-25796R1

Dear Dr. Anger,

We’re pleased to inform you that your manuscript has been judged scientifically suitable for publication and will be formally accepted for publication once it meets all outstanding technical requirements.

Kind regards,

Dzintars Gotham

Academic Editor

PLOS ONE

Additional Editor Comments (optional):

Reviewers' comments:

Reviewer's Responses to Questions

**Comments to the Author**

1. If the authors have adequately addressed your comments raised in a previous round of review and you feel that this manuscript is now acceptable for publication, you may indicate that here to bypass the “Comments to the Author” section, enter your conflict of interest statement in the “Confidential to Editor” section, and submit your "Accept" recommendation.

Reviewer #1: All comments have been addressed

Reviewer #2: All comments have been addressed

2. Is the manuscript technically sound, and do the data support the conclusions?

Reviewer #1: Yes

Reviewer #2: (No Response)

3. Has the statistical analysis been performed appropriately and rigorously? 

Reviewer #1: Yes

Reviewer #2: (No Response)

4. Have the authors made all data underlying the findings in their manuscript fully available?

Reviewer #1: Yes

Reviewer #2: (No Response)

5. Is the manuscript presented in an intelligible fashion and written in standard English?

Reviewer #1: Yes

Reviewer #2: (No Response)

6. Review Comments to the Author

Reviewer #1: All comments have been sufficiently addressed. I have no further remarks.

Reviewer #2: (No Response)

7. PLOS authors have the option to publish the peer review history of their article (what does this mean?). If published, this will include your full peer review and any attached files.

Reviewer #1: No

Reviewer #2: No

---

## [Editor Report · Acceptance letter]

31 Jan 2024

PONE-D-23-25796R1 

PLOS ONE

Dear Dr. Anger, 

I'm pleased to inform you that your manuscript has been deemed suitable for publication in PLOS ONE. Congratulations! Your manuscript is now being handed over to our production team.

Kind regards, 

on behalf of

Dr. Dzintars Gotham 

Academic Editor

PLOS ONE